# Temporal pattern recognition in retinal ganglion cells is mediated by dynamical inhibitory synapses

Simone Ebert [1,2,3,5] ✉, Thomas Buffet[3,5], B.Semihcan Sermet[3,4], Olivier Marre [3,6] & Bruno Cessac [1,2,6]

A fundamental task for the brain is to generate predictions of future sensory inputs, and signal errors in these predictions. Many neurons have been shown to signal omitted stimuli during periodic stimulation, even in the retina. However, the mechanisms of this error signaling are unclear. Here we show that depressing inhibitory synapses shape the timing of the response to an omitted stimulus in the retina. While ganglion cells, the retinal output, responded to an omitted flash with a constant latency over many frequencies of the flash sequence, we found that this was not the case once inhibition was blocked. We built a simple circuit model and showed that depressing inhibitory synapses were a necessary component to reproduce our experimental findings. A new prediction of our model is that the accuracy of the constant latency requires a sufficient amount of flashes in the stimulus, which we could confirm experimentally. Depressing inhibitory synapses could thus be a key component to generate the predictive responses observed in the retina, and potentially in many brain areas.

A long-standing hypothesis is that visual neurons do not signal the visual scene per se, but rather surprising events, e.g., mismatches between observation and expectation formed by previous inputs[1]. It has been observed in a number of sensory modalities that neurons strongly respond when a sequence of repetitive stimuli is unexpectedly interrupted[2–4].

In the retina, this phenomenon has been coined the Omitted Stimulus Response (OSR)[5]. When a periodic sequence of flashes suddenly ends, some ganglion cells emit a large response. Interestingly, the latency of this response shifts with the period of the flash sequence, so that the ganglion cell responds to the omitted flash with a constant latency. This suggests that the retina forms predictions of observed patterns, and responds to a violation of its internal expectation. Several studies have proposed potential mechanisms that may underlie this phenomenon, such as oscillatory activity in bipolar cells[6],

summation of parallel pathways with different response polarities[7] or response kinetics that respond early and late[8]. The fact that there is a rebound response at the end of the flash sequence can be simply explained by a model with a biphasic filter[7,8]. Previous work has shown that this rebound response depends on the ON bipolar cell pathway[9]. However, how the latency changes with the period of the sequence, such that the response to the Omitted stimulus has a constant latency, remains unclear. None of the mechanisms proposed so far could be experimentally proven. Thus, the mechanisms by which the retina achieves this latency shift remain unclear and debated.

Here we investigated how inhibitory amacrine cells affect the OSR and showed that depression in inhibitory synapses can account for this characteristic latency shift. To this end, we performed electrophysiological recordings of retinal ganglion cells and found that blocking inhibitory transmission from glycinergic amacrine cells

[1]INRIA Biovision Team, Université Côte d'Azur, Valbonne, France. [2]Institute for Modeling in Neuroscience and Cognition (NeuroMod), Université Côte d'Azur, Nice, France. [3]Sorbonne Université, INSERM, CNRS, Institut De La Vision, Paris, France. [4]Present address: Netherlands Institute for Neuroscience, Amsterdam, The Netherlands. [5]These authors contributed equally: Simone Ebert, Thomas Buffet. [6]These authors jointly supervised this work: Olivier Marre, Bruno Cessac. ✉e-mail: simoneebert166@gmail.com

selectively abolished the predictive latency shift of the OSR. We then showed that the latency shift is specific to the periodicity of the stimulus and is not a consequence of stimulus duration or luminance. To better understand how glycinergic inhibition impacts the latency of the OSR, we developed a circuit model equipped with a glycinergic amacrine cell. This model reproduced the latency shift of the OSR when the glycinergic synapse showed short-term depression, thereby adjusting its weight to the stimulus frequency. In addition, our model generated several predictions about the OSR, which we could confirm in experiments. The latency shift that is characteristic of the OSR is thus due to a depressing inhibitory synapse whose weight is changed by the stimulus frequency. For low-frequency sequences, the synaptic weight is large and this increases the latency of the response, while for

high-frequency stimuli, the weight is low due to depression, and the latency is only shifted by a small amount. Our results suggest a generic circuit to generate responses to surprise that could be potentially implemented in several brain areas.

## Results

### ON ganglion cells exhibit an omitted stimulus response to dark flashes

Using a multi-electrode array of 252 electrodes, we extracellularly recorded the spiking activity of ganglion cells from the mouse retina[10] (Fig. 1A). We presented sequences of 12 full-field dark flashes of 40 ms duration each, at frequencies of 6, 8, 10, 12, and 16 Hz, with a gray baseline illumination. We estimated the receptive fields of the same

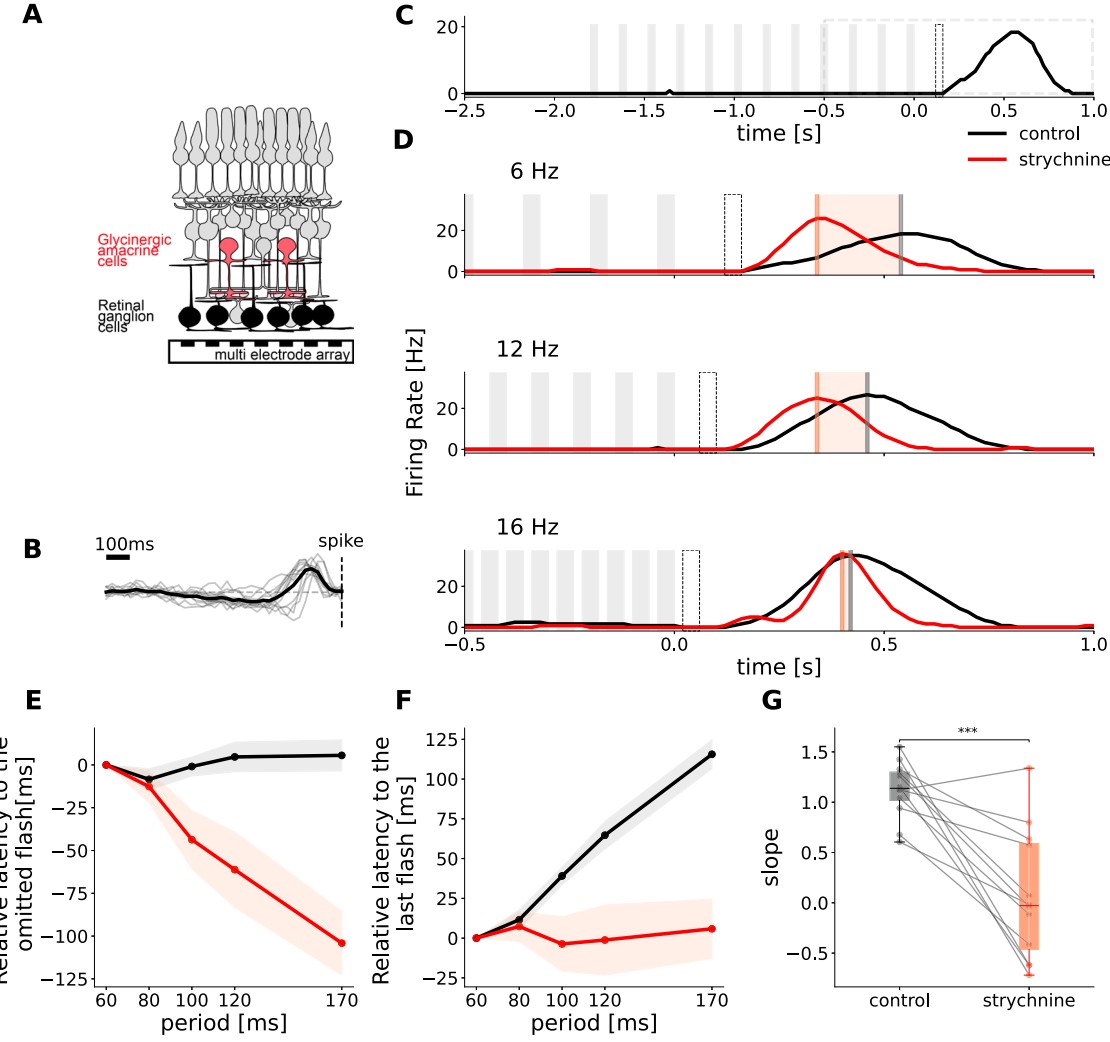

**Fig. 1 | Glycinergic amacrine cells are necessary for predictive timing of the OSR. A** Schematic representation of the retina, with the activity of retinal ganglion cells being recorded with a multi-electrode array. **B** Temporal traces of the receptive fields of the cells that loose the latency shift with strychnine, *n* = 12. Bold trace shows mean of all cells. **C** Example of OSR. The cell responds to the end on the stimulation, after the last flash. The times of dark flashes are represented by gray shaded rectangles. The black dotted rectangle shows the timing of the omitted flash. The black dotted rectangle shows the time period of focus in (**D**). **D** Experimental recording of the OSR in one cell in control condition (black) and with strychnine to block glycinergic amacrine cell transmission (red). Firing rate responses to flash trains of 3 different frequencies are aligned to the last flash of each sequence. Flashes are represented by gray patches, vertical lines indicate the maximum of the response peak, red shaded areas indicate the temporal discrepancy between control and strychnine conditions. **E** Mean ± SEM latency

between OSR and the omitted flash plotted against the period of the stimulus for a population of *n* = 12 cells. Latency is expressed relatively to the latency of the response to the 16 Hz stimulus in control condition. **F** Mean ± SEM latency between OSR and last flash in the stimulus plotted against stimulus period for a population of *n* = 12 cells. Control latencies shift with the period of the stimulus with a slope 1.13 ± 0.08. With strychnine, this shift is abolished (slope = 0.07 ± 0.18). Latency is expressed relatively to the latency of the response to the 16 Hz stimulus in the control condition. **G** Slope of the latency shift illustrated as a boxplot (band indicates the median, box indicates the first and third quartiles and whiskers indicate ±1.5 interquartile range). Circles show individual data points, lines connect values of the same cell. The slope decreases significantly when strychnine is added, $p = 1.09e^{-4}$ (two-sided Welch t-test). Source data are provided as a Source Data file. **A** is used with permission from Giulia Spampinato.

retinal ganglion cells with a checkerboard-like noise. We defined ON cells based on their receptive fields (responding to a light increase, Fig. 1B). Previous studies observed an OSR in both ON and OFF retinal ganglion cells. Given that previous pharmacological experiments revealed that ON inputs are necessary to drive an OSR[5], we focused on the response of ON cells to sequences of dark flashes.

Many ON cells responded with a broad peak of activity after the stimulus stopped for all frequencies tested ($n = 143$ cells). A subset of cells with response after stimulus end shifted the response latency according to the stimulus frequency (27 %, $n = 40$, Fig. 1D, see Methods). They exhibited an "Omitted Stimulus Response" (OSR), as it was described in[5]: when the period of the flash train increases, the latency of the response to the last flash in the stimulus shifts by the same amount (Fig. 1C, D), such that the latency of the response to the omitted stimulus is constant (Fig. 1E). This indicates that a subset of retinal ganglion cells have a temporal expectation of when the next flash should have occurred and shift their response latency accordingly. This is illustrated in Fig. 1F, where the relation between the latency of the response and the period of the flash sequence is linear, with a slope of nearly 1 (Fig. 1G). In the following we will refer to this specific relation as "latency shift".

To see whether there are ON cells showing this latency shift can be related to a known RGC type, we clustered them according to their response to a full-field "chirp" stimulation, which reveals response polarity, kinetics and preference to temporal frequencies and contrasts[11]. We identified roughly 4 different response types which we classified as "ON-transient", "ON-transient-sustained", "ON-sustained" and "ON-OFF" (see Fig. S1). This suggests that the OSR is not a response behavior specific to a given cell type but rather exhibited by a broader range of cells. While some cells showed a response to the onset of the flash sequence and/or showed small responses to flashes, the majority of cells did not respond at all to the flashed stimuli (see Fig. S1C). Additionally, the total latency between the last flash and the OSR varied substantially between cells, ranging between 142 and 483 ms for the fastest frequency (see Fig. S2, right). Even though the OSRs occur late for some cells, the latency difference of 120 ms between the fastest and the lowest frequency is very close to the difference in period (110 ms) of these stimuli. For further analysis, we chose to focus on the OSR peak characterized by this latency shift, pooling together ON cells of different types.

## Amacrine cells are required for the latency shift in the omitted stimulus response

It remains unclear how the retinal circuit generates an OSR. Previous studies have shown that the ON bipolar cell pathway is necessary to have a response per se[5], but the components of the retinal circuit needed for the latency shift are yet to be determined. We hypothesized that the inhibitory cells are responsible for the shift in latency, as inhibition has been shown to shift latency in various neuronal circuits[12–15]. Amacrine cells are the main class of inhibitory interneurons in the mouse retina. To investigate this further, we blocked glycinergic transmission using strychnine (2 μM) and recorded the spike responses of retinal ganglion cells to flash trains of varying frequencies (see Methods). As glycinergic transmission is only employed by certain classes of inhibitory amacrine cells in the mouse retina[16] this blocks only a subset of amacrine cells. In the following, we analyze the effect of strychnine on those ON retinal ganglion cells which exhibited a OSR with latency shift in control conditions.

While the response after the sequence end remained after strychnine application, we observed that the slope between response latency and stimulus period decreased from an average of $1.13 \pm 0.08$ in the control condition to $0.07 \pm 0.18$ after strychnine was added (Fig. 1F, G, mean ± SEM, $n = 12$, see Methods). While the OSR occurred at roughly the same time in the highest frequency tested, the peak was

significantly advanced after low-frequency flashes compared to the control condition (Fig. 1D). As a consequence, the OSR peak did not have a constant latency relative to the omitted stimulus after strychnine was added (Fig. 1E). These results demonstrate that glycinergic amacrine cells are a key contributor to the OSR latency shift. Although they do not generate the response alone, they are crucial for the peak latency shift, which is a hallmark of the OSR. We also looked at the onset time of the OSR response but found that the relation between onset time and stimulus period is highly variable with an average slope of 0.39 (see Fig. S3). The effect of glycine application on the onset of the OSR is also highly variable, resulting in a non-significant difference in the slope in control and strychnine conditions. Overall, these results suggest that spiking onset does not contain predictive information about the arrival of the next flash as its timing does not shift in a 1-to-1 manner with the period of the flash train.

## Stimulus luminance has no effect on response latency to opposite polarity stimuli

Previous works have shown that overall changes in luminance or contrast can induce temporal adaptation that could affect ganglion cell responses[17,18]. To further understand the mechanisms of the OSR, and constrain a model that would reproduce the experimental results, we investigated whether the latency shift of the OSR is really triggered only by the periodicity of the stimulus or if it is affected by other stimulus features. Periodic flash trains of different frequencies are not only characterized by periodicity but also vary in terms of average luminance and duration. As flashes have the same duration of 40 ms across all frequencies but occur in different windows of time, the average illumination and stimulus duration is lower for high than for low frequencies.

To investigate whether the latency shift of the OSR is simply a consequence of duration or different levels of average luminance, we presented two sets of stimuli: non-flashing dark step-stimuli of different lengths corresponding to the duration of flash trains presented in the control experiments, and non-flashing dark step-stimuli with different luminance levels corresponding to the average luminance during flash trains (Fig. 2A, B upper panel). Both stimuli evoked an offset response after stimulus in cells that show an OSR to periodic flashes (Fig. 2A, B lower panel), but the latency of this response was nearly equal across all luminance levels and durations tested (Fig. 2C). This suggests that the latency shift in the OSR is specific to periodic stimuli and that luminance or duration variations are not sufficient.

We then tested two further modifications of our periodic flash trains in which mean luminance was kept constant across stimulus frequencies (Fig. 2D). The first modification was to keep the luminance steady by increasing light intensity in between the dark 40-ms flashes, which yielded an OSR with latency shift (Fig. 2E, F, G, in blue). This suggests that overall contrast change is not necessary for the latency shift. In the second modification, we kept the average luminance constant by adjusting the duration of the flashes to half of the stimulus period. Surprisingly, we found that changing the duration of the flashes maintains the OSR peak but erases the latency scaling (Fig. 2E, F, G, in green). These experimental results suggest that the retina detects periodic patterns but can only form temporal predictions for specific input patterns (see also ref. 9).

Surprisingly, the responses to the stimulus with increased light intensity between the flashes were almost identical to the responses in the control stimulus. To both the control stimulus and the modification, only 4 out of 20 ON cells with OSR showed a response during the stimulus, mostly for low frequencies. We did not identify any quantifiable differences in the OSR response in these two cells compared to the remaining 16 cells that did not respond during the stimulus. Our hypothesis is that the transient increases are either too fast (for high frequencies) or too small (for low frequencies), to evoke a response above threshold.

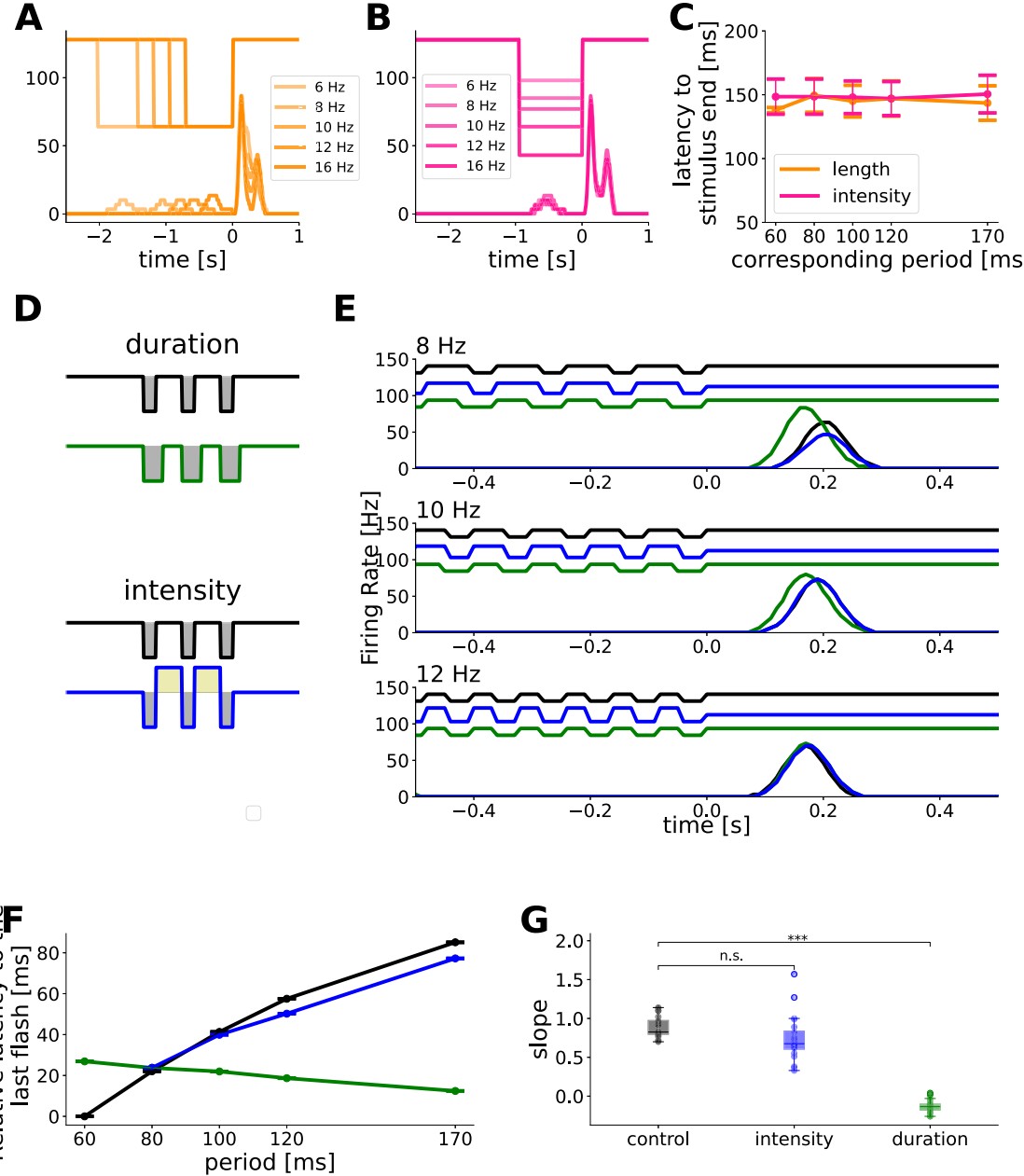

**Fig. 2 | Latency scaling of the OSR is selective to specific input patterns. A** Firing rate response of one example cell to dark step stimuli with different durations equivalent to the duration of flash trains with frequencies 6–16 Hz. **B** Firing rate responses of one example cell to dark step stimuli with different light intensities equivalent to the mean light intensity of flash trains with frequencies 6–16 Hz. **C** Mean ± SEM between response latency and corresponding stimulus period for step stimuli with duration or luminance modulation comparable to the stimulus composed of periodic flashes, $n = 20$. **D** Schematic description of the stimulus modifications for constant luminance across frequencies. "Duration" describes a modified stimulus in which the duration of each flash is set to half of the stimulus period. "Intensity" describes a stimulus modification in which the intervals between flashes are of brighter intensity to maintain a constant average luminance level.

**E** Response traces to flash trains of 3 different frequencies of each modification shown in (**D**). **F** Mean ± SEM scaling between response latency and stimulus period for all conditions, $n = 20$ cells. **G** Fitted slope between peak latency and stimulus period for $n = 20$ cells, illustrated as a boxplot (the band indicates the median, the box indicates the first and third quartiles and the whiskers indicate ±1.5 interquartile range). Black/blue/green circles show individual data points. Control mean ± SEM 0.89 ± 0.03, intensity mean ± SEM 0.74 ± 0.06, duration mean ± SEM −0.13 ± 0.01 Stimuli with intensity modulation lead to no significant change in slope ($p = 5.22e^{-2}$, two-sided Welch t-test) while duration modulations decrease the slope significantly ($p = 7.2e^{-23}$, two-sided Welch t-test). Source data are provided as a Source Data file.

## A circuit model with depressing synapses in inhibitory glycinergic amacrine cells explains the latency shift

Our experimental results provided compelling evidence that glycinergic amacrine cells provide inputs which are required to achieve the latency shift of the OSR and that the latency shift is specific to some stimulus patterns. However, it remained unclear how the retinal circuit achieves such selectivity in a glycinergic-dependent way. We

developed a mechanistic model in which we explicitly simulated evoked inputs from glycinergic amacrine cells to understand their role in the latency shift of the OSR.

We focused on ON ganglion cells and equipped our model with two ON inputs, one being excitatory $E^{ON}$ and mimicking ON bipolar cell input, and one being inhibitory $I^{ON}$, conveying broad delayed inhibition. This generates a biphasic response profile needed for a rebound

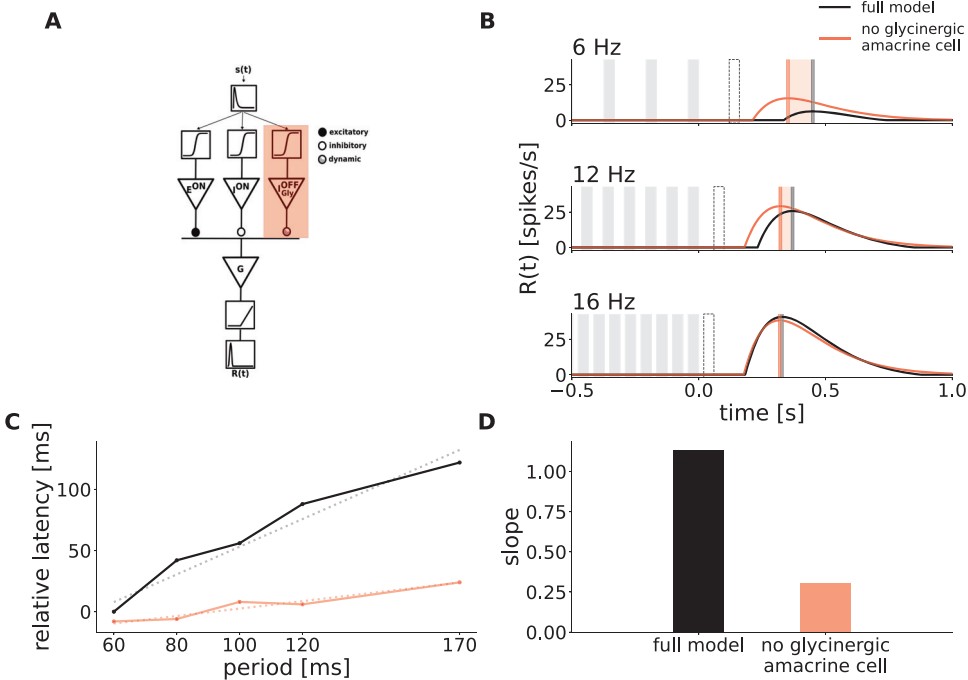

**Fig. 3 | Mechanistic model replicates latency shift and strychnine experiment.**
**A** Schematic description of the Model. It is composed of an ON excitatory input $E^{ON}$, an ON inhibitory input $I^{ON}$, and an OFF inhibitory input $I^{OFF}_{Gly}$ representing a glycinergic amacrine cell. Each of those units receives as input the convolution of the stimulus with a monophasic temporal kernel, determining the cells polarity, and connects onto a ganglion cell G. The response of G is then passed through a non-linearity to simulate the cells' firing rate. The synapse from $I^{OFF}_{Gly}$ to G can adapt its strength to the stimulus via short-term depression. The shaded red area represents the weight of this glycinergic amacrine cell being set to zero to simulate the effect of strychnine. **B** Simulation of the model responses to flash trains of 3 different

frequencies with the full model (black, control), and the weight of $I^{OFF}_{Gly}$ set to 0 (red, strychnine). The weight of $I^{ON}$ was changed from -65 to -36 Hz in this simulation, accounting for the broad effect of strychnine, which likely reduces inhibition overall. The times of dark flashes are represented by gray shaded rectangles, while the black dotted rectangle shows the timing of the omitted flash. **C** Latency of the OSR plotted against stimulus period in control and strychnine simulation. The latency is expressed relatively to the latency of the response of the full model to the 16 Hz stimulus. The slope of the latency shift decreased from 1.13 to 0.30 when $I^{OFF}_{Gly}$ is set to 0 (dotted lines). **D** Value of the slope fitted to latency shift in the full model and strychnine simulation. Source data are provided as a Source Data file.

response to dark flashes[7]. This delayed inhibitory input summarizes the influence of various inhibitory pathways (horizontal cells, GABAergic amacrine cells, ON glycinergic amacrine cells) that generate the biphasic response profile (see Discussion). In addition, we explicitly included a glycinergic amacrine cell with an OFF polarity, $I^{OFF}_{Gly}$, in order to provide inhibition to dark stimuli. All three units receive inputs from the outer plexiform layer (OPL), which is passed through a nonlinearity to simulate nonlinear processing in the first synaptic retinal layer. All 3 units synapse onto a ganglion cell G, where synaptic inputs to G are again passed through a nonlinearity (Fig. 3A, see Methods for details).

A characteristic feature of the OSR is that the latency shifts by the same amount as the stimulus period. In other contexts, it has been shown that the relative strengths of excitation and inhibition can determine the latency of the response[12–15]. We reasoned that the effective strength of the inhibitory input should thus depend on the frequency of the flash sequence. This can be achieved with a dynamic synapse, i.e., a synapse whose strength varies with the frequency of the stimulation.

Several previous reports have shown that inhibitory synapses can be depressing, i.e., have a decreasing weight depending on previous inputs[19–23]. It has been hypothesized that the variation in synaptic strength results from varying availability of vesicles in the readily releasable vesicle pool, which gets gradually depleted upon persistent inputs[17,24,25]. In the retina, this has been modeled via dynamical systems of vesicle pools, where the value of one of the variables directly serves as input to the postsynaptic cell[26,27]. To keep our model as simple as possible, we modeled the glycinergic synapse as a depressing synapse using one kinetic equation to simulate synaptic vesicle occupancy,

similar to cortical models of short-term plasticity[28–30] but replaced spiking inputs by continuous presynaptic voltage. The vesicle occupancy then scales the output of the cell (see Methods).

This mechanistic model allowed us to reproduce the main properties of the recorded ganglion cells. Thanks to excitatory and inhibitory ON inputs, it has an ON biphasic impulse response. It also responds with a peak at the end of dark periodic flash stimuli, due to delayed disinhibition, because inhibition has a slower temporal filter than excitation (Fig. 3B).

Our model also successfully simulated an OSR whose latency with respect to the last flash increased with the stimulus period with a slope around 1, and thus with a constant latency with respect to the omitted stimulus (Fig. 3B, C, black line).

We next simulated the experimental effect of strychnine with the model by removing the glycinergic amacrine cell input. Since the ON inhibitory cell of our model also includes the effect of ON glycinergic amacrine cells, we simultaneously decreased the weight of the ON inhibitory input (note however that our results did not depend on that additional modification, see Discussion). Fig. 3B, C (red) show that the model replicates the experimental effect of strychnine. The slope of the latency shift decreases from 1.16 to 0.34 when $I^{OFF}_{Gly}$ is removed from the circuit.

To test whether the model achieves the latency shift of the response peak thanks to the dynamical synapse, we simulated the response of the model while keeping the occupancy of the glycinergic amacrine cell synapse constant at 1 (Fig. 4A). The latency increased in all frequencies simulated, but the slope of the latency shift decreased to 0.32 (see Fig. 4B–D). Without the dynamical synapse, the linear filters in the distinct pathways of our model can be summed together

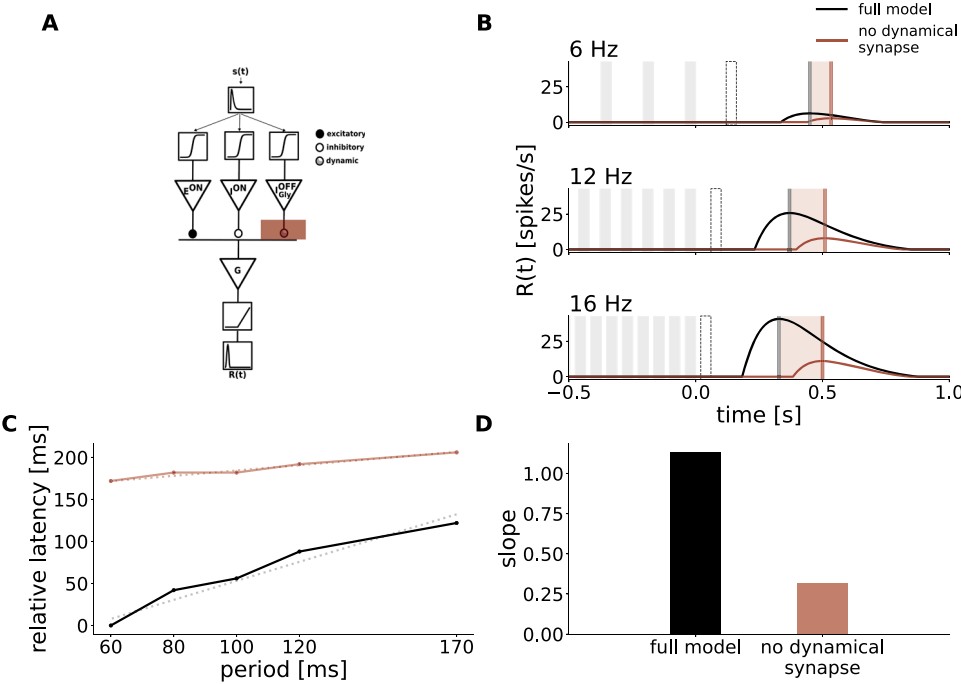

**Fig. 4 | Short-term plasticity is the crucial component for latency shift.**
**A** Schematic description of the Model, as in Fig. 3. The shaded red area now represents removing the dynamical characteristic of the glycinergic synapse. **B** Simulation of the model responses to flash trains of 3 different frequencies with dynamic occupancy (black) and the weight of $I_{Gly}^{OFF}$ held constant (red). The times of dark flashes are represented by gray shaded rectangles, while the black dotted rectangle shows the timing of the omitted flash, as in Fig. 3. **C** Latency of the OSR

plotted against stimulus period in control and strychnine simulation. The latency is expressed relatively to the latency of the response to the 16 Hz stimulus in the control condition. The slope of the latency shift decreased from 1.13 to 0.31 when the weight of $I_{Gly}^{OFF}$ was held constant (dotted lines). **D** Value of the slope fitted to latency shift in the full model and without the adaptive property of the glycinergic synapse. Source data are provided as a Source Data file.

to one ON biphasic filter profile. Followed by the non-linearity placed in the ganglion cell our model is equivalent to a LN-model and very similar to the one proposed by ref. 7. However, these models, which are somewhat simplification of ours (they have no dynamical synapse) are not able to predict the latency shift with a slope of 1. The dynamical synapse is thus essential to achieve the latency shift of the response with a slope of 1 observed experimentally. Note that there was also an overall increase in latency which can be explained by the fact that the fixed synapse is stronger than the dynamic counterpart since it is never at full occupancy when stimulated due to depression.

Additionally, we observed that the OSR amplitude decreased when stimulus frequency increased (Fig. S4B). There are no significant changes in this relation after strychnine application, albeit the curve seems slightly flattened. Our model captures this relation accurately (Fig. S4A).

Finally, we simulated the model's response to modified flash trains with constant average luminance across periods (Fig. 2D). As observed in experiments, the model maintains an OSR with latency shift to periodic flash trains with increased light intensity in inter flash-intervals but loses this scaling in response to flash trains where the duration of the flashes is adjusted (see Fig. S5).

The model thus captures experimental findings accurately.

### The depressing inhibitory synapse induces a latency shift
To understand how the model can account for the latency shift in the OSR, it is helpful to look at the temporal evolution of its internal variables. The ON excitatory input evokes a hyperpolarization in response to dark flashes and cancels the depolarization evoked by the slightly slower ON inhibitory cells during the flash sequence. At the end of the flash sequence, due to differing time constants, there is a time window where depolarization due to the inhibitory delayed cell exceeds the hyperpolarization due to the bipolar cell, and

triggers a spiking response of the retinal ganglion cell (Fig. 5A). This is similar to many classical rebound responses recorded experimentally, and it can be predicted with a biphasic filter. But by itself, this biphasic filter would not predict the latency shift as observed experimentally (see above). As we will describe in the following paragraph, this latency shift can be explained thanks to the specific effect of the glycinergic amacrine cell equipped with a depressing synapse.

Since this glycinergic amacrine cell is an OFF cell, it inhibits the ganglion cell in response to dark flashes. This has the effect of delaying the spiking response at the end of the flash sequence and to increase its latency (Fig. 5B, black compared to gray). The latency of the response is then shifted for different stimulus frequencies because the depressing synapse changes the strength of the glycinergic inhibition and the time scale in the response.

In this synapse, the vesicle occupancy represents the amount of synaptic depression and decreases when stimulation starts (Fig. 6A, 3rd row). It then reduces the current input from $I_{Gly}^{OFF}$ to the ganglion cell (Fig. 6A, 4th row). This reduction of inhibition shifts the OSR towards an earlier response, reducing its latency (Fig. 6A, 5th row).

Fast-frequency stimuli cause stronger depression, reducing the $I_{Gly}^{OFF}$ current input by about 30 %. This has a strong impact on the latency, which is more than 100 ms shorter when the synapse is depressed. In contrast, slow-frequency stimuli cause only weak depression, reducing $I_{Gly}^{OFF}$ by about only 10 %. The latency was thus only slightly reduced in that case (compare Fig. 6A, B).

In summary, the steady-state vesicle occupancy of the synapse is determined by the stimulus frequency (Fig. 6C, D). The vesicle occupancy can then reduce the inhibitory current, yielding a large reduction for fast inputs and a small reduction for slow inputs (Fig. 6E). As a result, vesicle occupancy acts as a scaling factor to tune the inhibitory current input and thereby shifts the response latency based on

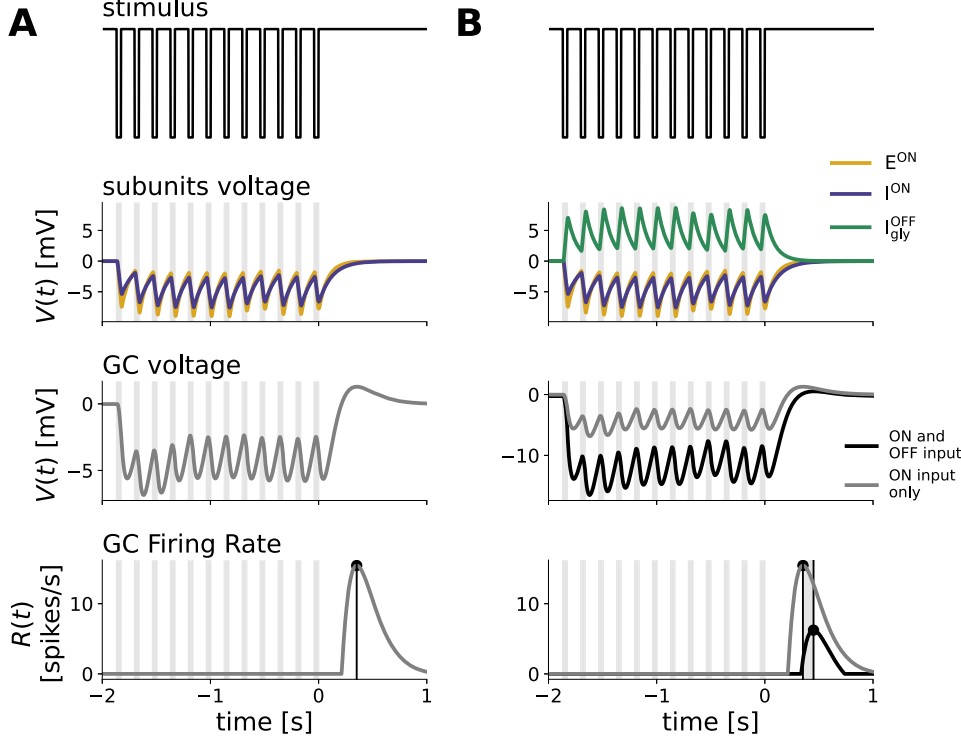

**Fig. 5 | ON components of the model produce a peak after stimulus end while the glycinergic OFF input shifts the latency.** From top to bottom: Stimulus intensity, bipolar and amacrine voltage responses, ganglion cell voltage and firing rate. **A** Internal model responses of ON components to a 6 Hz dark flash stimulus without OFF inhibition. ON excitation and inhibition hyperpolarize in response to dark flashes. When both inputs are substracted in the ganglion cell, its voltage sum hyperpolarizes during flash presentation, followed by an overshoot of disinhibition due to the slower response profile of the inhibitory input. After passing the voltage

through a rectification function, only the disinhibitory peak after stimulus end remains in the firing rate. **B** Effect of additional OFF inhibition. The voltage of the glycinergic OFF input depolarizes in response to dark flashes, passing additional inhibition onto the ganglion cell. This lowers the GC voltage response and increases the latency between peak and stimulus end in the firing rate. Last two panels compare the models' simulation and peak time-point with (black) and without (gray) the input from $I_{Gly}^{OFF}$. Source data are provided as a Source Data file.

stimulus frequency. This explains how the latency shift observed experimentally is achieved via glycinergic inputs.

### The depressing inhibitory synapse predicts other features of the omitted stimulus response

Can our model give other predictions about the OSR? Since the glycinergic inhibitory synapse is more depressed at high frequency, the OSR is less inhibited, and its amplitude is thus stronger compared to low frequencies (compare Fig. 6A, B, 5th row). This trend was also observed in our experiments. Both simulated and experimental amplitudes showed a negative correlation between the response amplitude and the stimulus period ($-0.87 \pm 0.02$, mean $\pm$ SEM $n = 14$) in data and as well in simulations ($-0.93$).

An important consequence of our depressing synapse model is that it needs several flashes to reach the steady state in the vesicle occupancy. If we shorten the flash sequence, the vesicle occupancy will barely reach that steady state, and this should have predictable consequences on response amplitude and latency. We simulated the response to long flash trains consisting of 12 flashes (as in the experiments and simulations above) and shorter sequences of only 5 flashes.

Our simulations predicted that the amplitude of the OSR decreases when the stimulus contains only 5 flashes. We tested that in experiments and found the same tendency: the OSR amplitude was smaller in all but the lowest frequency tested (see Fig. 7A).

Another model prediction was that the slope of the relation between OSR latency and stimulus period should decrease for shorter flash trains, reaching only a value of 0.67 for 5 flashes, compared to 1.16 when 12 flashes were presented (see Fig. 7B, left). In our model, this is a consequence of the dynamics of the depressing synapse.

In the 5-flashes scenario, our model predicts that the stimulus is too short for the synapse to reach a reliable steady-state occupancy that is not perturbed by the oscillations due to the flashes. (Fig. 7C). This perturbs the response latency scaling and changes the slope of the relation between latency shift and stimulus period (Fig. 7D) for short flash trains.

In our experiments, while there was no difference for low-frequency stimuli, the absolute latency of the OSR was much larger after 5 flashes than after 12 when the stimulus frequency was high (see Fig. 7B). This change in latency led to a reduction of the mean $\pm$ SEM slope value from $0.84 \pm 0.02$ to $0.69 \pm 0.04$ in experiments, consistent with the model prediction. These agreements provide further evidence for the validity of our model, and for the key role of a depressing inhibitory synapse.

## Discussion

The OSR is an example of sophisticated feature detection that takes place already in the retina. This phenomenon implies that retinal ganglion cells can carry a dynamic prediction of their future visual input with high temporal precision, and selectively respond when this prediction is not matched (Fig. 8A). Although high-contrast full-field periodic flashes are artificial stimuli that are unlikely to occur in natural scenes, they isolate temporal aspects of visual input patterns. The underlying mechanisms of temporal processing in the rather artificial OSR may be embedded in more complex networks detecting spatio-temporal patterns in more realistic scenes.

With this work, we provide evidence that the latency shift of the OSR, which allows a constant latency relative to the omitted stimulus, is generated by inhibition from glycinergic amacrine cells and

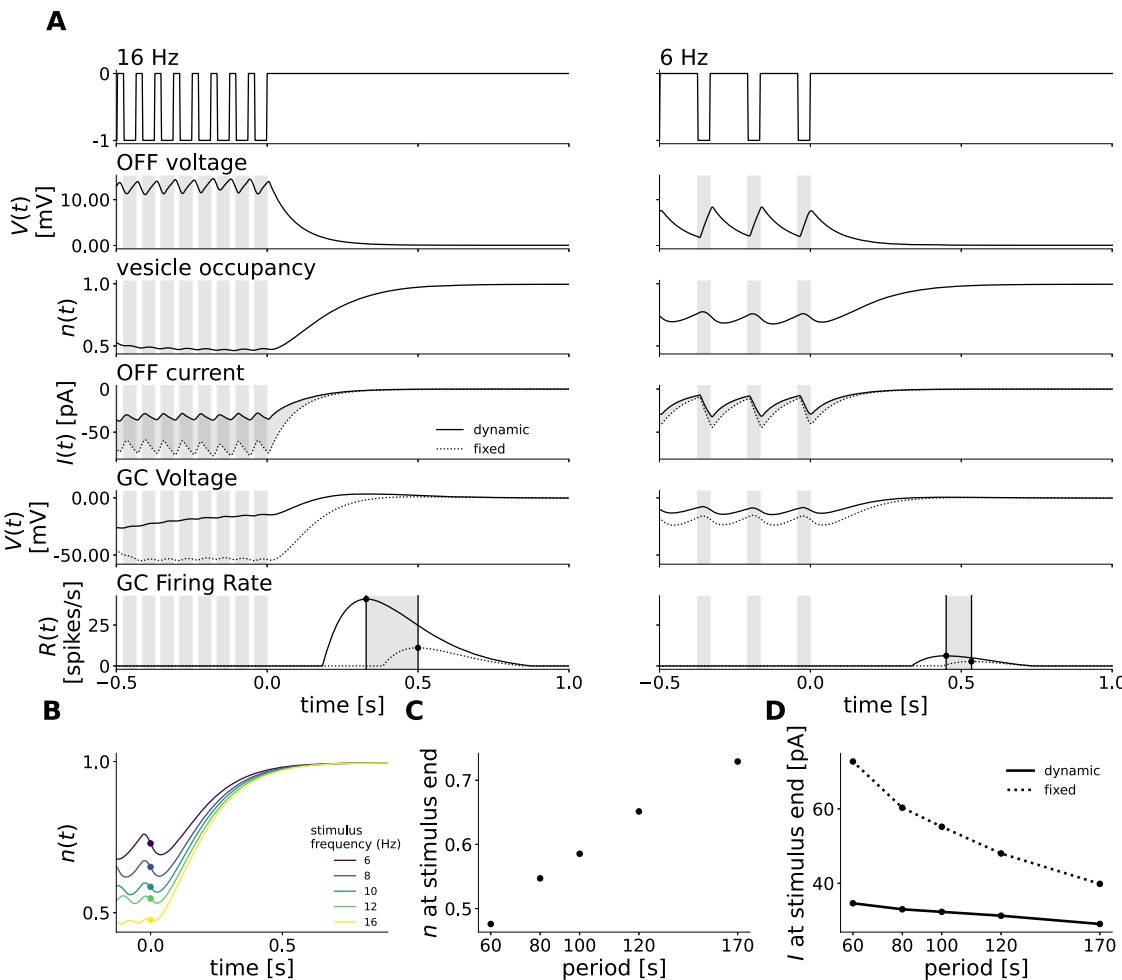

**Fig. 6 | Synaptic depression scales OFF glycinergic input to stimulus frequency and thereby shifts the latency of the response.** Responses shown are zoomed in around the end of the flash sequence. **A** Impact of occupancy scaling on $I_{Gly}^{OFF}$ current input to G for a fast (16 Hz, left) and a slow (6 Hz, right) stimulus. From top to bottom: Stimulus intensity, $I_{Gly}^{OFF}$ voltage, vesicle occupancy, current input, G voltage, and firing rate. Last 3 panels compare simulations with dynamic occupancy (solid lines) to when the occupancy is held constant (dotted lines). Depression has the effect to advance the OSR peak, more so for fast than slow frequencies. Current of unit X is computed with equation (5) in Methods. **B** Occupancy traces to flash stimuli of different frequencies aligned to the last flash. Dots indicate the occupancy level at stimulus end. **C** Level of occupancy after stimulus end scales with the period of the stimulus. **D** $I_{Gly}^{OFF}$ current input is decreased by short-term depression, more so for fast than slow frequencies. Dotted line shows current with fixed occupancy, solid line with dynamic occupancy. Source data are provided as a Source Data file.

triggered by the periodicity of the stimulus. Using computational modeling, we show how inhibition can enable retinal ganglion cells to respond to the missing flash at the end of a sequence. Short-term depression in inhibitory synapses allows shifting the latency of this response (Fig. 8B, C).

Several theoretical models have been proposed to elucidate the mechanisms behind the OSR. Werner and Passaglia[7] proposed a dual LN-model with biphasic ON-OFF pathway interactions, which accurately captures the response peak after the stimulus ends via the rebound phase of the pathway selective to the opposite polarity than the stimulus. However, it fails to shift the peak latency as a function of the stimulus period with a slope of 1, which is a defining feature of the OSR. This slope value is necessary to have a response of constant latency with respect to the omitted stimulus. When removing the depressing synapse, our model is amenable to a biphasic LN-model, since the responses of our intermediate units are then linear and could be represented by a single linear filter as well. Our model then simulates the OSR in the same manner as this previous study, but the depressing inhibitory synapse was necessary to obtain the slope of 1, which is the signature of a predictive latency shift.

Gao and Berry[6] proposed intrinsic oscillatory activity in ON bipolar cells that evoked a latency shift via resonance tuned to the stimulus frequency. However, such oscillatory activity was not found in bipolar cells[31]. Further, our experiments show that glycinergic amacrine inhibition is necessary for the latency scaling of the OSR, which cannot be explained by this model.

More recently, ref. 8 proposed that the OSR with its latency shift can arise in a deep neural network model via summation of multiple excitatory inputs with different time constants. It is difficult to evaluate whether the model accurately captures the latency shift as observed in experiments, with the correct slope value. The explanation behind this model is that the OSR latency is determined by the sum of 2 ON bipolar cells which are activated only by certain stimulus frequencies due to different temporal filtering. This purely excitatory mechanism of latency scaling is not in line with our experimental findings, suggesting that amacrine cells likely contribute to temporal filtering as well. Our hypothesis is thus that the components they isolated correspond to a mix of bipolar and amacrine cell properties.

In contrast to those previous models of the OSR, we explicitly included an inhibitory input whose contribution to the peak latency is dependent on the stimulus frequency via short-term plasticity. By

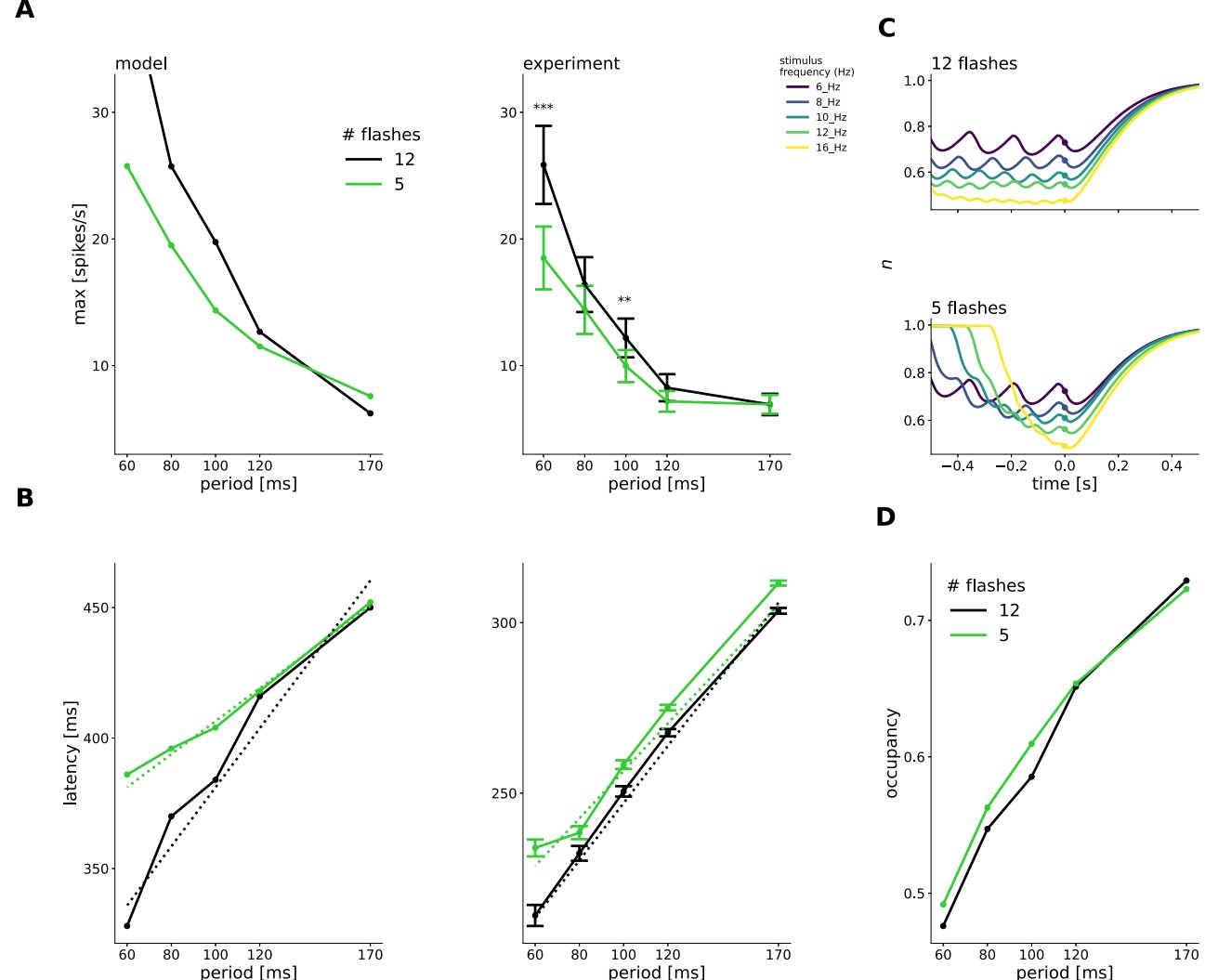

**Fig. 7 | Latency shift decreases for shorter stimuli because of lacking steady state occupancy. A** Mean ± SEM amplitude of the OSR against stimulus period for 12 and 5 flashes in the stimulus in simulations (left) and experiments (right). Amplitudes to 6 Hz and 10 Hz stimuli were significantly different according to two-sided t-test with Bonferroni-Holm correction[47] (16 Hz: $p = 0.000006$, 10 Hz: $p = 0.001$). **B** Latency against stimulus period in simulations (left) and experiments (right). Simulated slopes decreased from 1.13 after 12 flashes to 0.63 after 5 flashes in simulations. Experimental slopes decreased from 0.84 ± 0.02 to 0.69 ± 0.04,

mean ± SEM. The latency after 12 and 5 flashes was not significantly different according to two-sided t-test with Bonferroni-Holm correction[47] (6 Hz: $p = 0.2$, 8 Hz: 0.14, 10 Hz: $p = 0.044$, 12 Hz: $p = 0.031$, 16 Hz: 0.024). **C** Temporal traces of vesicle occupancy to all frequencies simulated, for 12 flashes (upper) and 5 flashes (lower). Dots indicate occupancy at stimulus end. Traces do not reach a steady state for 5 flashes. **D** Scaling of occupancy with stimulus period in 5- and 12-flashes scenario. Source data are provided as a Source Data file.

doing so, we can propose a mechanistic explanation and match the latency shift of the OSR as well as various other response properties of the experimentally observed OSR.

Previous experimental studies[5,7,9] reported that the OSR is found in a higher proportion of retinal ganglion cells than what we experimentally observed in this study. This difference could come from the fact that we define OSR as a response with a latency shift having a slope of at least 0.7, while it is not clear whether previous studies took multiple frequencies into account when classifying the OSR. Schwartz et al.[9] also showed that blocking inhibition from amacrine cells had no effect on the OSR. But again, this study only investigated the presence or absence of the OSR under amacrine blockade, and did not investigate if the latency of the OSR shifted with the stimulus frequency. In addition, some previous studies were mostly carried out in salamander[5,9], where the underlying mechanisms may be different from the mouse.

In order to realistically simulate the model's response with glycinergic amacrine cells blocked, we had to decrease the weight of the

inhibitory ON input $I_{ON}$ in our simulations. Leaving the weight of this input untouched while setting $w_{I_{Gly}^{OFF}}$ to 0, we still obtain a decrease in latency shift but this configuration generated a response to each flash of the sequence, something we did not observe experimentally. We therefore deemed this configuration as less realistic than decreasing also the weight of the ON inhibitory cell, since strychnine is likely to affect glycinergic ON inhibition as well. Additionally, the components of our circuit might represent several cell types pooled together, and more detailed circuit models might give similar predictions. For example, we chose to simulate synaptic depression via a modified version of cortical STP-models with only 2 parameters rather than the more complex systems used in the retina previously[27]. Overall, we chose to only include the minimal components necessary to specifically explain the peak latency shift in the OSR and its abolishment via strychnine and arrived at a model with (still) 19 parameters.

In addition, bath application of strychnine acts on the whole retinal network and likely has side effects such as changes (increases) in the baseline activity across bipolar, other amacrine, and ganglion cells

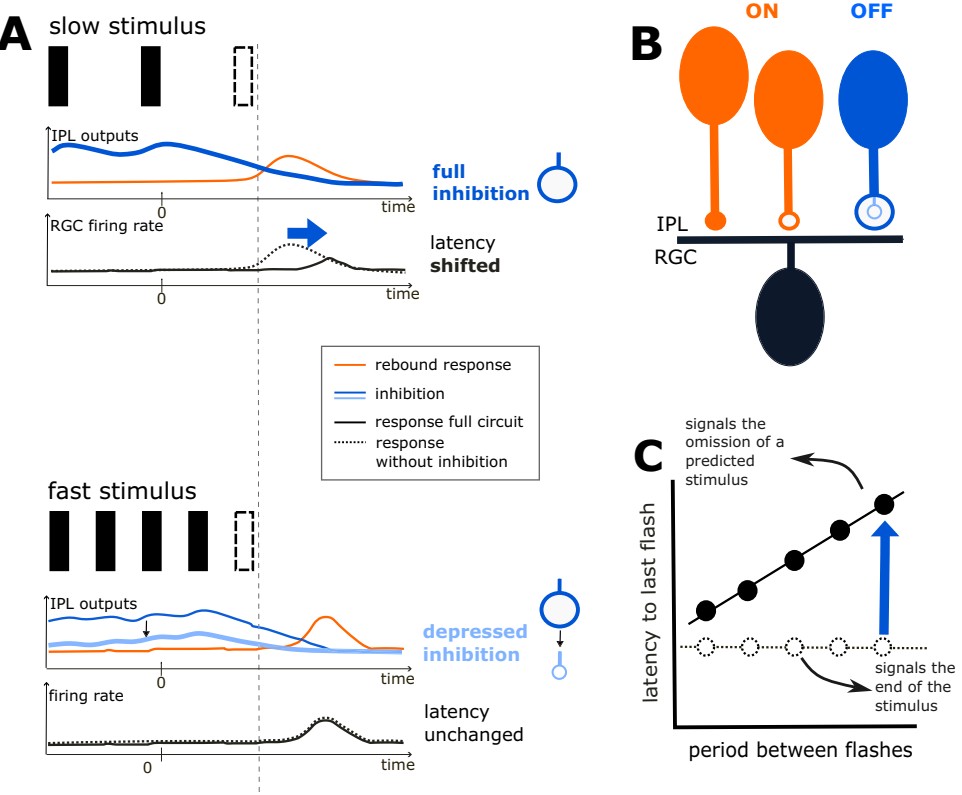

**Fig. 8 | Mechanistic neuronal circuit explains the latency shift of the Omitted Stimulus Response (OSR) via depressing inhibition. A** Schematic responses of components shown in (**B**) to a slow-frequency and a fast-frequency stimulus. Black rectangles represent dark flashes in a periodic sequence. The dotted rectangle represents an omitted flash at the end of the sequence. For both stimulus speeds, the upper panel shows inputs from the IPL. ON (orange) inputs are combined into one trace and OFF inhibitory input is shown in blue. ON inputs generate a peak response after stimulus ends. OFF inhibitory input is activated by the dark flash sequence and then slowly decreases after stimulus ends, overlapping with the ON peak. The lower panel shows the RGC firing rate. Without OFF inhibition (dotted black), the RGC emits a peak at the same time as the ON inputs. Adding OFF inhibitory inputs (solid black) generally decreases the this response and push the peak backward. The impact of inhibition depends on the stimulus frequency due to synaptic depression. At slow-frequency inputs, synaptic depression is not activated. Inhibitory inputs remain strong and shift the latency of the OSR (blue arrow) to a later time. Fast-frequency stimuli trigger strong synaptic depression, leading to a depletion of inhibitory inputs. The latency remains unchanged. **B** Schematic description of the circuit. A retinal ganglion cell (RGC, in black), which emits an OSR with latency shift, receives 3 inputs in the IPL (inner plexiform layer). Orange units are ON inputs, which provide excitation (filled circle) and inhibition (empty circle). The blue unit provides OFF inhibition and has a dynamical synapse which depresses after activation. **C** Latency of the OSR to the last flash in the stimulus plotted against stimulus period. A RGC without depressing inhibitory inputs (dotted black) will emit a response at roughly the same time after stimulus ends. It thus merely signals the end of the stimulus. With depressing inhibition (solid black), the latency is shifted backward for slower frequencies (with longer periods), which leads to a to a 1-to-1 scaling between latency and stimulus period. Via this mechanism, the latency of the OSR with respect to the omitted flash remains constant across frequencies, indicating that the RGC signals the omitted stimulus. The latency can thus provide a prediction of the time when the flash should have occurred.

due to a reduction of maintained inhibition. However, these maintained effects, although they would likely decrease absolute latencies, should not depend on the stimulus and are thus unlikely to explain changes in the latency shift relative to stimulus period. For simplicity, we thus do not take these side effects into account in our modeling approach and only simulate the minimal amount of direct inputs we found necessary for an OSR with latency shift.

Our present model accounts only for an OSR to dark-type flashes in ON cells. It has however been reported as well that OFF cells emit a similar response after cessation of OFF-type stimulation which our model as it is does not explain. This could be obtained by adding another independent OFF pathway that would deliver an input to the same ganglion cell. This pathway would trigger a response during the flashes, while the circuit described in our model would generate an OSR that scales with latency.

We experimentally identified an important role for glycinergic inhibition in the latency shift of the OSR in experiments. The role of GABAergic inhibition in generating the OSR is less clear. In principle, depressing GABAergic synapses could play a similar role as glycinergic synapses or be the source of disinhibition needed for the rebound response. We tried to block GABAergic inputs with Gabazine to test this hypothesis experimentally. We indeed found a small number of cells that also lost their latency scaling and others that lost the entire OSR after application of gabazine (data not shown), supporting both hypotheses. However, we observed a large heterogeneity in the effect of gabazine on the OSR across experiments, so these results remain inconclusive (see Fig. S7).

In this study, we focused on the OSR defined as the response to the abrupt cessation of a periodic stimulus, following several previous studies[5–9]. It has however been shown that retinal ganglion cells in both mouse and salamander emit an OSR to more complex pattern violations, for example in response to an omission in the middle of a periodic stimulus sequence[5], which we did not test here. We would expect to observe such a response if the overall latency of the OSR for a given frequency was shorter than twice the stimulus period. In a stimulus where one flash is omitted in the middle of a sequence, this would allow for the OSR to signal this omission before flash sequence continues. According to absolute latencies of cells recorded in this study, several cells should signal omissions in the middle of a flash stimulus up to frequencies of 12 Hz. (see Fig. S7).

Dynamical synapses have previously been proposed to enable neuronal circuits in the retina to form expectations of future inputs[32,33] and are thus a plausible candidate to play an important role in the OSR. Previous works have shown that inhibitory synapses can be depressing[21,34,35]. In particular, glycinergic synapses that input to bipolar cells can be depressing[23]. However, there is no method to experimentally remove the depressing nature of the synapse without affecting the inhibitory weight, so we could not show experimentally that the depressing nature of the synapse is necessary for the OSR.

Adaptation in excitatory ribbon synapses is a well-studied phenomenon to reliably encode luminance and contrast and could be the source of the OSR latency shift as well. While a more simple model with depression in bipolar cell terminals can provide an OSR with latency shift as well (see Fig. S6), such a model is not congruent with our findings that stimulus luminance and contrast are neither necessary nor sufficient for the OSR latency shift and cannot readily explain our experimental observations under strychnine. While it has been shown that GABAergic inhibition modulates gain changes in ribbon synapses[18], glycinergic effects at this stage have not been observed so far. Therefore we conclude that, even though plasticity in bipolar cells certainly plays an important role in retinal computations, the idea of inhibitory plasticity provides a simpler explanation of this specific phenomenon.

In addition, our model predicts that the depressing inhibitory synapse should have several functional consequences, that we verified in the data. In particular, a key prediction of the depressing synapse is that the OSR requires a long enough flash sequence to accurately shift the latency and we confirmed this prediction experimentally.

Ultimately, our results might be of relevance to understanding neuronal mechanisms of predictive coding beyond the retina. Very similar surprise responses exist in other sensory domains, such as the mismatch negativity response in the auditory cortex[2,36–38], where neural activity is enhanced following a "deviant" tone in a sequence of "standard" tones. A recent study suggested that synaptic adaptation could be a key contributor to this phenomenon[39]. Following the predictive coding theory, one possible explanation is that this response emerges from an interaction between feed-forward and feedback connectivity[40,41].

Here we show that a purely feed-forward micro-circuit can generate this response to a violation of prediction via an interplay of excitation and inhibition, where synaptic depression takes place in inhibitory connections. All the components used in this micro-circuit are generic and can be found in other sensory areas[28,42,43], and it is thus likely that a similar circuit could be at work at the cortical level, for more complex pattern recognition than full-field flashes.

## Methods

### Experimental setup
**Recordings.** Recordings were performed on isolated retinas from 8 C57BL6/J adult mice aged between 2 and 7 months. Strychnine Experiments were performed on two males (age 84 and 52 days) and one female (85 days). Luminance experiments were performed on two retinas from one female (161 days). Gabazine experiments were performed on two females (58 and 163 days) and one male (185 days). Experiment with different flash numbers was performed on a female mouse (30 days). The animals were housed in enriched cages with ad libitum food, and watering. The ambient temperature was between 22 and 25 °C, the humidity was between 50 and 70% and the light cycle was 12–14 h of light, 10–12 h of darkness. Animals were killed according to institutional animal care standards of Sorbonne Université. The retina was isolated from the eye under dim illumination and transferred as quickly as possible into oxygenated Ames medium (Merck, A1420). The retina was extracted from the eye cup and lowered with the ganglion cell side against a multi-electrode array whose electrodes

were spaced by 30 µm[10]. During the recordings, the Ames' medium temperature was maintained at 37 °C. Raw voltage traces were digitized and stored for off-line analysis using a 252-channel preamplifier (MultiChannel Systems, Germany) at a sampling frequency of 20 kHz. The activity of single neurons was obtained using Spyking Circus V 1.1.0, a custom spike sorting software developed specifically for these arrays[44].

**Visual stimulation.** Visual stimuli were presented using a white LED and a Digital Mirror Device. Flash sequences contained 5 or 12 flashes of 5 different frequencies (6 Hz, 8 Hz, 10 Hz, 12 Hz, 16 Hz). Polarities were either switched from gray to black (dark flashes) or from gray to white (bright flashes). 60 trials were conducted for each stimulus, with 2–4 s between each trial. The order of magnitude of the background illumination was $10^6$ R*.

**Spike triggered averages.** We displayed a random binary checkerboard during 40 min to 1 h at 40 Hz to map the receptive fields of ganglion cells. A three-dimensional STA (x, y, and time) was sampled averaging over the stimulus preceding each spike for a time window of 1 s, divided into $N = 40$ time bins. Temporal and spatial components were isolated via Singular Value decomposiion and reconstructed from Eigenvectors. STA analysis was carried out in Python.

**Pharmacology.** To block glycinergic transmission, we dissolved strychnine (Sigma-Aldrich, S8753) in Ames' medium at a concentration of 2 µM, and perfused the retina with the solution at least 15 min before the recording.

**Latency analysis.** To determine slope of latency shift, we measured the latency between the peak firing rate and the end of the last flash in the stimulus for all frequencies tested. We plotted these latencies against the respective period of the stimulus and fitted a straight line to determine the slope of the latency shift using SciPy's optimization package[45]. Cells were classified as having an OSR in the control condition when the slope was at least 0.7 or higher. All cells where the peak time point could not be unambiguously determined in any condition were excluded from the analysis.

**Statistical analysis.** Statistical testing was performed using Scipy's stats package[45]. Welch's t-test for independent samples with unequal variance was used for slope comparisons between conditions. For comparison of amplitudes and latencies between different flash numbers, paried two-sided t-tests with Bonferroni-Holm correction were uses such that significance levels $\alpha$ were adjusted to $\frac{\alpha}{m+1-k}$ for the $k^{th}$ $p$ value of $m$ total comparisons. Pearson correlation coefficients were calculated using Python's numpy build-in functions.

### Modeling
**Model implementation.** The 3 pathways of Fig. 3 receive an input from the Outer Plexiform Layer (OPL) written as a temporal convolution of the OSR stimulus, $s(t)$ (dimensionless), with a linear filter of the form:

$$\alpha(t) = \frac{t}{\tau_{OPL}^2} \exp\left(-\frac{t}{\tau_{OPL}}\right) H(t), \tag{1}$$

where $\tau_{OPL}$ is the characteristic time of integration (in $s$) in photoreceptors and $H(t)$ the Heaviside function.

Thus, the output of the OPL reads:

$$F(t) = [\alpha * s](t), \tag{2}$$

where $*$ is the space-time convolution. Note that, the stimulus being spatially uniform, the space integration reduces to a constant, so that the detailed shape of the spatial RF plays a trivial role here.

The OPL response is then passed through a sigmoidal nonlinearity to account for nonlinear processing steps in early bipolar cell processing, given by:

$$\sigma_{ab}(F) = \frac{S_X}{1 + e^{-a(F-b)}} \qquad (3)$$

where $a$ and $b$ are parameters defining the shape of the nonlinearity (they depend on $X$ but we didn't add this dependence to alleviate notations). $|S_X| = \frac{S_{mv}}{\tau_X}$ is a scale factor which scales the input into each unit to $\frac{mV}{s}$ via $S_{mV} = 20mV$ and the time constant $\tau_X$ of each unit, such that the amplitude of the response is normalized to the time constant of the respective unit. For OFF cells, the OPL inputs are multiplied by -1 before rectification to reverse polarity. The rectified signal is then integrated into each pathway via a linear dynamical system:

$$\frac{dV_X}{dt} = -\frac{V_X}{\tau_X} + \sigma_{ab}(F(t)), \quad X = E^{ON}, I^{ON}, I^{OFF}_{Gly}. \qquad (4)$$

where $V_X$ is the voltage of cell X (in Volt) and $\tau_X$ its characteristic time constant (in s). The Current $I_X$ of unit X is computed as

$$I_X = \frac{w_X \, p_\theta(V_X)}{c_m} \qquad (5)$$

with a membrane capacitance set to $c_m = 0.1nF$.

Note that this dynamical system is non autonomous as $F(t)$, the input explicitly depends on time.

Next, all pathways provide input to the ganglion cell G:

$$\frac{dV_G}{dt} = -\frac{V_G}{\tau_G} + w_{E^{ON}} p_\theta(V_{E^{ON}}) + n w_{I^{OFF}_{Gly}} p_\theta\left(V_{I^{OFF}_{Gly}}\right) + w_{I^{ON}} p_\theta(V_{I^{ON}}). \qquad (6)$$

where $w_{E^{ON}}, w_{I^{OFF}_{Gly}}, w_{I^{ON}}$ are synaptic weights (in Hz). Voltages are rectified before integrated in the ganglion cell membrane potential via :

$$p_\theta(V) = \begin{cases} V - \theta & \text{if } V \geq \theta : \\ 0 & \text{otherwise} . \end{cases} \qquad (7)$$

where $\theta$ is a threshold (in V) that differs between pathways.

The synaptic weight from $I^{OFF}_{Gly}$ to G is modulated by a dimensionless variable $n$, used to simulate synaptic short-term plasticity. $n$, which interprets as a vesicle occupancy in the glycinergic amacrine synapse, obeys the kinetic equation[30] :

$$\frac{dn}{dt} = (1 - n)k_{rec} - \beta k_{rel} p\left(V_{I^{OFF}_{Gly}}, \theta_{I^{OFF}_{gly}}\right) n. \qquad (8)$$

$k_{rec}$ and $k_{rel}$ are rate constants (in Hz) for vesicle release and replenishment and $\beta$ (in $V^{-1}$) is a scaling factor. Finally, the voltage response is passed through the piece-wise linear function $p$ to obtain the firing rate:

$$R(t) = s_G \, p(V_G(t), \theta_G), \qquad (9)$$

where $s_G$ is a scaling factor. The model was implemented with custom Python code.

**Parameter optimization.** First, time constants and weights were fitted such that the linear response $V_G$ of the model (without nonlinearities) to an impulse stimulus best resembles the temporal STA of one example cell with OSR. This was done using an evolutionary optimization algorithm using the CMAES (Covariance Matrix Adaptation Evolutionary Strategy) Python toolbox[46]. Synaptic plasticity is removed from the model by setting $\beta = 0$. The slope of the ganglion

**Table 1 | Model parameter values used in simulations**

| Parameter | Value | Unit |
|---|---|---|
| $\tau_{OPL}$ | 0.003 | s |
| $\tau_{E^{ON}}$ | 0.08 | s |
| $\tau_{I^{ON}}$ | 0.085 | s |
| $\tau_{I^{OFF}_{Gly}}$ | 0.12 | s |
| $\tau_G$ | 0.11 | s |
| $a_{ON}$ | 14.0 | 1 |
| $a_{OFF}$ | 12.0 | 1 |
| $b_{ON}$ | −0.5 | 1 |
| $b_{OFF}$ | 0.5 | 1 |
| $w_{E^{ON}}$ | 50.0 | Hz |
| $w_{I^{ON}}$ | −65.0/36.0 | Hz |
| $w_{I^{OFF}_{Gly}}$ | −53.0/0.0 | Hz |
| $S_{E^{ON}}$ | 250.0 | mV s$^{-1}$ |
| $S_{I^{OFF}_{Gly}}$ | −235.3 | mV s$^{-1}$ |
| $S_{I^{ON}}$ | 166.7 | mV s$^{-1}$ |
| $\theta_{E^{ON}}$ | 26.0 | mV |
| $\theta_{I^{ON}}$ | −20.0 | mV |
| $\theta_{I^{OFF}_{Gly}}$ | 0.0 | mV |
| $k_{rel}$ | 5.0 | Hz |
| $k_{rec}$ | 10.0 | Hz |
| $\beta$ | 0.0826 | m V$^{-1}$ |
| $\theta_G$ | 0.0 | V |
| $s_G$ | 2200 | Hz mV$^{-1}$ |

cell non-linearity $S_G$ was then chosen as to match experimentally observed firing rates. Parameters of the occupancy equation (8) were manually chosen as to yield a slope of 1. Thresholds for synaptic rectifications were calculated such that the rest potential of G remained at 0 without input. Slopes and thresholds for nonlinear processing within units were manually chosen. Values were obtained by screening over a wide range of combinations of $\beta$ and the ratio of recovery and release rate $\frac{k_{rec}}{k_{rel}}$. All parameters used in the simulations are listed in Table 1.

**Reporting summary**
Further information on research design is available in the Nature Portfolio Reporting Summary linked to this article.

**Data availability**
The data generated in this study have been deposited in the Zenodo database under access code https://doi.org/10.5281/zenodo.11396185. Due to large file sizes, raw datasets are available from the corresponding author upon request. Source data for all Figures are provided with this paper. Source data are provided with this paper.

**Code availability**
The code used for the computational model is accessible on gitlab.

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

## Acknowledgements

We thank Romain Brette, Matthias Henning, and Romain Veltz for helpful discussions about the model, and Matias Goldin and Brice Bathellier for critical reading of the manuscript. We thank Giulia Spampinato for allowing us to use her excellent artwork. This work has been funded by a PhD fellowship from the interdisciplinary Institute for Modeling in Neuroscience and Cognition (NeuroMod) of Université Côte d'Azur to S.E., funded by the National Research Agency (ANR-15- IDEX-01) and, by an ERC grant (No 101045253, DEEPRETINA) to O.M., ANR grants (DECORE, ANR-18-CE37-0011, and PerBaCo, ANR-22-CE37-0016-02) to O.M., a grant from Retina France to O.M., and ANR ShootingStar (ANR-20-CE37-0018-04) to O.M. and B.C. T.B. was funded by a PhD fellowship from ENS and supported by the Fondation pour la Recherche Médicale, grant number FDT202304016465.

## Author contributions

S.E., T.B., O.M. and B.C. designed the study. T.B. and B.S.S. performed the experiments. S.E. and T.B. analyzed the data with help from O.M. and B.C. S.E. did the model with help from T.B., O.M. and B.C. S.E., T.B., O.M. and B.C. wrote the paper.

## Competing interests

The authors declare no competing interests.
