## [Peer Review File · Nature Communications]

Temporal pattern recognition in retinal ganglion cells is mediated by dynamical inhibitory synapsesREVIEWER COMMENTS

Reviewer #1 (Remarks to the Author):

In this study, the authors examine the contribution of depressing inhibition to the omitted stimulus response using a combination of pharmacology and modeling. Overall, the results are interesting in their mechanistic explanation, but there are several areas in which the manuscript could be improved that I outline below.

Major concerns:

I struggle with the general description of the putative omitted stimulus response (OSR) for several reasons:

- 1) Only one example response of the phenomenon in a cell shown in the paper. Are there different response patterns to the stimulus as shown in the original Schwartz et al paper.
- 2) The example that is shown in Figure 1C,D show that the cell's peak response occurs ~500 msec after the omitted flash should have occurred. It is hard to reconcile this observation with the idea the phenomenon is somehow indicating that the expected stimulus did not occur and providing something informative about the timing of when that omitted stimulus should have occurred.
- 3) Related to point 2 is the idea the timing of the cellular responses are linearly related to the temporal frequency of the stimulation, as presented in panels E/F of Figure 1. However, in panel D of the same figure, the peak responses are all occurring between 400-500 ms after the offset of the last flash. This seems problematic because it would be difficult to distinguish this type of behavior (general response at the offset of repeated stimulation due to disinhibition) from the purported pacemaker-like OSR.
- 4) Making strong conclusions about the mechanism based on the pharmacology experiments seems difficult. Bath applying strychnine will block both maintained and evoked glycinergic amacrine cell input. The interpretation of the results in the text is based on blocking the evoked portion of amacrine cell input. The maintained portion of this block will change the activity (i.e., V_{rest}) across bipolar cells and ganglion cells and this can have many unintended consequences such as changing the input-output relationship and general excitability. This should at least be acknowledged in the text. Further, the authors conclude that the phenomenon is mediated by glycinergic amacrine cells, but the putative OSR phenomenon doesn't go away with strychnine -- the kinetics just speed up, which is expected when the excitatory cells are pushed closer to threshold (maintained effect).
- 5) Why did the authors not entertain the possibility that GABAergic amacrine cells were also involved? This should be clarified in the text.
- 6) In order to observe the OSR behavior, the On cells in this paper are presented with repeated dark flashes (i.e., opposite to the preferred polarity). What would be the prediction for a model in which the cone bipolar cells had biphasic temporal filters and showed their typical nonlinear input-output relationships. How much of this behavior would one expect to see under these conditions (i.e., a model with no inhibition and a reasonable spike threshold for the ganglion cell). This seems like it would be the null-hypothesis in sorting out the putative contribution of amacrine cells.
- 7) As mentioned in (6), the OSR is observed when the cell is presented with repeated, high-contrast, non-preferred stimuli. These stimulus statistics are almost certainly never observed in the natural

environment. There should be some acknowledgment of this and some discussion of how/whether this behavior would ever be expected in natural vision.

8) The manuscript as a whole would benefit from more effort in placing the current work in the context of previous work on the OSR, starting in the Introduction.

Minor concerns:

1) In Figure 1B, it appears that there are at least two different modes for the temporal filters (transient vs sustained). How are the authors distinguishing cell type? It seems that there could be several types that show the same behavior.

Reviewer #2 (Remarks to the Author):

NCOMMS Temporal pattern recognition in retinal ganglion cells is mediated by dynamical inhibitory synapses

This is a very interesting paper and I really enjoyed reading it.

The paper deals with the Omitted Stimulus Response (OSR) in the retina, which is an example of a more general phenomenon in which sensory neurons respond unusually strongly when a repetitive stimulus is interrupted. In other, more general language, the OSR is an example of a "mismatch" signal highlighting the difference between observation and expectation based on the recent history of the sensory input. These types of computations are of general interest in many areas of neuroscience but we understand surprisingly little of the underlying circuit mechanisms.

In this paper the authors home in on the temporal properties of the OSR – the latency between the omitted stimulus and the response, which is dependent on the period of the repeated stimulus. Through a simple but powerful combination of experiment and modelling they provide good evidence for a key insight: depressing inhibitory synapses control the latency of the response to an omitted stimulus. In the retina, these are glycinergic synapses, but the model is quite generic and could be easily transposed to other neural circuits where GABAergic inhibition dominates. I therefore think that this work will have impact not just for the field of retinal processing but also more generally for those investigating the circuit basis of similar "mismatch" computations in other neural circuits. Short-term changes in the strength of both excitatory and inhibitory synapses (primarily depressing changes) are likely common to many (all?) circuits when they are driven hard enough and this paper provides a beautifully approachable example of how this interplay can control the timing of a mismatch response.

The work supports the conclusions well and I cannot see any major flaws in the analysis, modelling or interpretation.

Major points

1. My major suggestion for improving presentation is to describe more clearly the experimental observations and model predictions for the amplitude of the OSR. These results are important to give an overview of the role of glycinergic inhibition and are briefly mentioned on p. 12 but I would have liked to have seen a more thorough presentation, both because the amplitude of responses is an essential aspect of the phenomenon being modelled and because the current literature is confusing on this point. I suggest that Fig. 1 include a panel showing how the amplitude of the OSR depends on

stimulus period and the effects of glycinergic block on that relation. The model outputs in Fig. 2B indicate that glycinergic block should increase OSR amplitude with a stronger relative effect at lower frequencies. There is a hint of a similar trend in the example experiments in Fig. 1D but a systematic analysis of the whole data set is needed. It may be that the model does not neatly account for changes in amplitude, and this would need discussion.

2. The abstract says: "Here we show that depressing inhibitory synapses enable the retina to signal an omitted stimulus in a flash sequence". I don't think the word "enable" is accurate in terms of what is shown. Blocking glycinergic inhibition does not block the OSR – it adjusts its timing. So reword to be more specific and accurate.

3. The authors do not investigate the role of GABAergic inhibition and readers outside the retina will wonder why this is not mentioned. Some words at some point about the difficulties of interpreting more generalized block of inhibition across the whole retinal circuit would help those readers understand in which an "omitted experiment response" is generated by the mismatch between their expectations and contents of the paper.

Minor points

1. the abstract says "We also generated new predictions with this model, that we confirmed experimentally.". It would be good to be more specific about the nature of these predictions (I assume the results in Fig. 5 and 6 are being referred to).

2. Section 2.5. para beginning "A characteristic feature in the Omitted Stimulus Response is that the latency shifts by the same amount as the stimulus period. It has been shown that the relative strengths of excitation and inhibition can determine the latency of the response [34, 37, 38, 8].". At first I thought the OSR in the retina was being referred to but checking the references the authors are discussing the OSR more generally. Would be helpful to add that in first sentence.

3. Discussion: sentence beginning "Dynamical synapses have previously been proposed to enable neuronal circuits in the retina to form expectations of future inputs [13] and are thus a plausible candidate to play an important role in the OSR". The role of excitatory and inhibitory synapses in generating the predictive responses described in ref. 13 (Hosoya et al. 2005) were investigated relatively directly in Johnson et al. 2019 (A Retinal Circuit Generating a Dynamic Predictive Code for Oriented Features. Neuron. 2019 Jun 19;102(6):1211-1222.e3).

4. Discussion could perhaps be improved with some subheadings for major topics.

Leon Lagnado

Reviewer #3 (Remarks to the Author):

The manuscript "Temporal pattern recognition in retinal ganglion cells is mediated by dynamical inhibitory synapses" proposes a new mechanism by which the retina may achieve a phenomenon called the Omitted Stimulus Response (OSR). This is a neural behavior previously described in which the retina responds in a special manner to the omission of a stimulus from a sequence of flashes (Schwartz et al. Nature Neuroscience 2007). Here, the authors explore the circuit configuration and properties that could produce this phenomenon. They use standard methods including (1) recording

the OSR in the mouse retina using a multi-electrode array to record local potentials of ganglion cells, (2) building a retina model that recapitulates the behavior of the experimentally recorded ganglion cells, and (3) testing whether the model predicts other aspects of how the retina responds to the flashing stimulus. The authors propose an interesting solution, which is that a depressing glycinergic synapse can produce the specific latency features of the OSR—a much less complicated and more biologically plausible solution than those previously proposed in the literature. However, my comments below center on an alternative possibility that has not been addressed and that would re-frame the OSR as a (possibly less interesting) contrast sensing phenomenon. The authors should rule this possibility out.

Overall, these results will be interesting to retinal neurobiologists interested in the role of inhibition in predictive coding the retina.

Major comments

1. It appears that OSR may be related to a more general phenomenon in the retina, which is the well-described phenomenon of contrast encoding. The dark flashes presented at different frequencies in Fig. 1D are of the same duration, leading to an overall average luminance that is lower during stimulation for the 16 Hz stimulus than during the 6 Hz stimulus. Thus, at the end of this stimulation period, the contrast change is greater for 16 Hz than 6 Hz. It is well-described that bipolar cell release encodes luminance and contrast through mechanisms related to vesicle release at the ribbon and inhibition onto the bipolar cells (ie. Oesch & Diamond 2019, Visual Neuroscience). In addition, the overall stimulation period is different length for each of these stimuli, and it is also established that the duration of suppression of ribbons in both photoreceptors and bipolar cells alters the kinetics and magnitude of their release (i.e. Jackman et al. 2009 Nature Neuroscience; Vlasits et al. 2014 Neuron). The longer that ribbon synapses are suppressed, the larger the glutamate discharge will be upon release from suppression due to an accumulation of vesicles. The fact that the authors do find that they must stimulate with a sufficient number of flashes (Fig 6, 5 vs. 12 flashes) to produce this phenomenon supports the idea that this phenomenon may be related to contrast encoding by ribbon synapses.

Thus, I am skeptical of the special/complicated nature of this phenomenon, and wonder whether it is simply a special case of the ganglion cells encoding contrast. This issue does not seem to have been addressed in the literature on the OSR.

To address this, the authors should show that the OSR response is specific to flashing stimuli and that the ganglion cells do not respond in the same manner to a non-flashing stimulus with the same average contrast over time. Another option would be to equalize the contrast change by varying the duration of the flashes in addition to their frequency. The authors should also demonstrate that bipolar cells with luminance/contrast encoding ribbon properties (as described in the literature mentioned above) are not sufficient to produce the OSR phenomenon in their model in the absence of the depressing glycinergic input.

2. While the peak response of the OSR appears to shift with different frequencies of stimulation, the spiking onset appears to begin at about the same time in control and strychnine conditions (Fig. 1D). This suggests that the ganglion cell does start firing at about the same time after the last flash, which is quite different from what is shown in the model (Fig. 2 B) where firing onset and peak both depend on the stimulus frequency in the full model. The authors should explain why peak firing rate is the relevant parameter for calculating the latency to omitted stimulus and why their model results do not match the experimental data in this respect.

Minor comments

1. The authors describe the ON ganglion cells as biphasic, but in Fig. 1B there appear to be many monophasic cells and a few biphasic cells. The authors should clarify the difference between the cells selected for analysis and other cell types recorded.
2. The authors should clarify the units in their modeling plots so it is easier to evaluate whether the values of these parameters are plausible (Fig. 4, 5 voltage and current traces)
3. Blue traces in Fig 5B and Fig. 6C are difficult to distinguish. The authors should make it easier to see the difference.
4. The slope of the latency shift appears to become much more variable in the presence of strychnine (Fig. 1G). This reviewer is curious if there are different effects in subpopulations of cells shown here. The authors should demonstrate/describe the details of this distribution by showing the population data with lines between the control and strychnine measurements. In addition, a statistical test that takes into account the unequal variances should be used.
5. The methods section should describe the computer software used for all analysis, modeling, and statistical testing.

Reviewer #4 (Remarks to the Author):

This paper aims at investigating the mechanisms behind the so-called Omitted Stimulus Responses (OSR) in retinal ganglion cells. This previously observed phenomenon describes the strong response peak in some ganglion cells when a sequence of periodic light stimuli suddenly ends or misses an individual stimulus in the sequence. In particular, when the period of the light stimulus sequence increases, the latency of the response peak increases in a corresponding fashion.

In the present manuscript, the authors combine experimental recordings from mouse retina and computational analyses to derive a circuit model that aims at explaining the OSR and its latency dependence on the stimulus period. As a key element, the authors propose that ON ganglion cells receive OFF inhibition via a synapse with short-term depression. This is an interesting model that is nicely illustrated and analyzed.

However, a substantial drawback of the presented material is that the investigated scenario does not correspond to the typical phenomenon of an OSR. The authors investigate responses of ON cells to a sequence of OFF flashes, whereas the classical investigations of the OSR in the retina focused on responses of OFF cells to such a sequence of OFF flashes (even though some examples of ON cells were also included). The fundamental difference is that, for an ON cell, the termination of the OFF flashes corresponds effectively to an increase in luminance and thus an activating stimulus. Adding a latency dependence to the resulting response seems not so mysterious and has not been, for example, the issue of interest in previous investigations of the mechanism of the OSR. For the classical OSR, on the other hand, one would need to explain how the response peak of an OFF cell, after cessation of OFF-type stimulation, is generated in the first place, in conjunction with its latency dependence on stimulus frequency.

In the present model, for example, the response seems not to depend on the periodic nature of the stimulus, and a continuous dimming would probably cause similar rebound activation, with latency likely depending on the amplitude of the dimming stimulus.

The implementation of the rebound is also somewhat confusing, as the manuscript states that the

voltages that form the input into the ganglion cell undergo a thresholding operation (Equation 5). Yet, the thresholds for the ON excitation and ON inhibition are not given in the table, and it seems that these inputs should be mostly linear (or have a very low threshold) for the rebound to occur. This is not impossible, but less plausible than rectifying synapses, and effectively, the ON inhibition then acts like an OFF excitation.

In addition, while the modeling of the depressing inhibitory input is interesting, the evidence of such a mechanism in the experimental data is not very strong. The block of glycinergic inhibition is a rather unspecific test, which likely throws off the excitation-inhibition balance and could thus disturb the latency dependence even if this inhibition is not directly involved in the mechanism. Also, the derived predictions (e.g. lower OSR amplitude for lower frequencies and shorter sequences) are not very specific for the depressing inhibitory input, as the predictions essentially state that a weaker OFF stimulus leads to a weaker rebound response. This can likely also be explained by nonlinear or depressing excitatory input.

Response to Reviewer Comments

Nature Communications manuscript NCOMMS-23-06244-T

March 2024

First of all, we would like to thank all reviewers for their constructive feedback and insightful remarks. We tried to thoroughly address all concerns raised and they helped improving our manuscript. We are grateful for the reviewers' comments and we think addressing their comments has made our manuscript substantially stronger. In response to important points raised, we performed additional experiments in which we

1. tested the effect of luminance and contrast changes on the latency scaling of the OSR and
2. a possible involvement of GABAergic transmission for the generation of the OSR.

As we performed additional experiments, the statistics presented in our first results section slightly changed (n of cells, % of cells with OSR) and are highlighted in **red**. We also modified our initial model by including nonlinear processing into our cellular units to account for the new results we obtained. The parameter, latencies and slopes of our model slightly changed, all changes are as well highlighted in **red**

In the following response, all remarks from the reviewer are stated and followed by a response to each remark (in *italic*). All changes in the manuscript are highlighted in **red** and referenced with a tag to the respective remark labelled **[RnPk]**, where n is the reviewer index and k the remark index.

Reviewer 1 (Remarks to the Author):

In this study, the authors examine the contribution of depressing inhibition to the omitted stimulus response using a combination of pharmacology and modeling. Overall, the results are interesting in their mechanistic explanation, but there are several areas in which the manuscript could be improved that I outline below.

Major concerns

I struggle with the general description of the putative omitted stimulus response (OSR) for several reasons:

Remark 1. Only one example response of the phenomenon in a cell shown in the paper. Are there different response patterns to the stimulus as shown in the original Schwartz et al paper ?

Response. In our experiments, some cells showed a response to the onset of the flash sequence and/or showed small responses to flashes for the 6Hz stimulus. However, the majority of cells did not respond at all to the flashed stimuli and we chose not to distinguish between OSR response types as reported in Schwartz et al, 2007 in recordings from the salamander retina. To our knowledge it is not clear if all these response patterns have been reported in the mouse as well. We include a new Supplementary Figure S1 (p.25) which shows all OSRs to 12Hz stimulation (panel C) and added a brief mention about this in the beginning of section 2.1, p.3.

Remark 2. The example that is shown in Figure 1C,D show that the cell's peak response occurs 500 msec after the omitted flash should have occurred. It is hard to reconcile this observation with the idea the phenomenon is somehow indicating that the expected stimulus did not occur and providing something informative about the timing of when that omitted stimulus should have occurred.

Response. We acknowledge that some of the the omitted stimulus responses we observe occur late, but we would like to point out that there is a substantial variety in the overall offsets after which the OSR responses begin. We included a Supplementary Figure S1 (p.25) which shows the OSR responses of all cells to 12 Hz stimulation, where the absolute latency is visible. The absolute latencies after 12 Hz stimulation range between 529 and 177 ms, with a median at 214 ms (see Figure 1). It might be that temporal information about stimulus history are relevant for retinal computations at multiple time scales. See also response to Remark 3.

Figure 1: *Absolute latency of the OSR after 12 Hz stimulation of $n = 35$ cells included in our study. Latencies range between 529 and 177 ms, with a median at 214 ms. Shades blue area shows the density, individual datapoints are shown in light blue.*

Remark 3. Related to point 2 is the idea the timing of the cellular responses are linearly related to the temporal frequency of the stimulation, as presented in panels E/F of Figure 1. However, in panel D of the same figure, the peak responses are all occurring between 400-500 ms after the offset of the last flash. This seems problematic because it would be difficult to distinguish this type of behavior (general response at the offset of repeated stimulation due to disinhibition) from the purported pacemaker-like OSR.

Response. Even though some of the omitted stimulus responses we observe occur late, there is a latency difference of 120 ms between fastest and lowest frequency, which is very close to the difference in period (110ms) of these stimuli. This is something we would not expect from an offset response due to dis-inhibition, that would occur with the same delay to all frequencies. We added a mention of this in Results section 2.1, p.3.

Remark 4. Making strong conclusions about the mechanism based on the pharmacology experiments seems difficult. Bath applying strychnine will block both maintained and evoked glycinergic amacrine cell input. The interpretation of the results in the text is based on blocking the evoked portion of amacrine cell input. The maintained portion of this block will change the activity (i.e., Vrest) across bipolar cells and ganglion cells and this can have many unintended consequences such as changing the input-output relationship and general excitability. This should at least be acknowledged in the text. Further, the authors conclude that the phenomenon is mediated by glycinergic amacrine cells, but the putative OSR phenomenon doesn't go away with strychnine – the kinetics just speed up, which is expected when the excitatory cells are pushed closer to threshold (maintained effect).

Response. We agree with the reviewer on the fact that strychnine can also

affect general excitability. For simplicity, we did not take any indirect side effects of strychnine bath application, such as baseline changes (increases) in bipolar, amacrine or ganglion cells into account in our modeling approach. We only simulated the minimal amount of direct inputs we found necessary and sufficient for an OSR with latency shift. We acknowledge that baseline activity changes likely affect response kinetics and thereby response latencies. However, these maintained effects, although they would likely decrease absolute latencies, should not depend on the stimulus and are thus unlikely to explain changes in the latency shift relative to stimulus period. Therefore, we conclude that evoked glycinergic inputs are likely responsible for the latency shift of the OSR, but not for the response peak per se. We included a statement about this in Discussion section 3.2, p.18.

Remark 5. Why did the authors not entertain the possibility that GABAergic amacrine cells were also involved? This should be clarified in the text.

Response. This is an important concern raised by the reviewer and led us to perform additional experiments with Gabazine bath application. We found that the effect of Gabazine on the OSR is quite heterogenous (see Figure 2). In some cells, Gabazine rarely affected the OSR (2A), while in others it led to a decrease in slope (2C) and yet in others, it eliminated the peak of the OSR entirely (2E). Our model indeed consists of 2 types of inhibition that serve distinct roles, one (glycinergic) unit that scales the response latency and a (possibly GABAergic) one that generates the OSR via disinhibition. Cells that loose the OSR after Gabazine application are supporting this idea. Nevertheless, the rebound could have other sources than disinhibition such as for example inputs from the OFF pathway or synaptic vesicle accumulation, which would explain why the OSR peak in other cells is not affected. Lastly, it seems that GABAergic inhibition might contribute to the latency scaling for some cells as well. However, due to the variety of effects observed, we decided not to include these results in the manuscript. We mention them in the Discussion 3.2, p.18-19.

Figure 2: *Gabazine has various effects on the OSR. A.,C.,E.* Response traces to flash trains of 3 different frequencies in control (black) and gabazine (shades of purple) conditions. *B.,D.,E.* Mean relative latency to the last flash of the stimulus plotted against stimulus period for control (black) and gabazine (shades of purple) conditions. Dotted line shows slope of 1. *A.,B* Example cell where OSR remains largely unaffected by gabazine. (Slope control = 0.71, gabazine = 0.4., $n = 2$) *C.,D* Example cell where OSR slope decreases with gabazine. (Slope control = 0.97, slope gabazine = 0.35, $n = 4$). *E.,F* Example cell where OSR disappears though gabazine. (Slope control = 0.96, $n = 4$).

Remark 6. In order to observe the OSR behavior, the On cells in this paper are presented with repeated dark flashes (i.e., opposite to the preferred polarity). What would be the prediction for a model in which the cone bipolar cells had biphasic temporal filters and showed their typical nonlinear input-output relationships. How much of this behavior would one expect to see under these conditions (i.e., a model with no inhibition and a reasonable spike threshold for the ganglion cell). This seems like it would be the null-hypothesis in sorting out the putative contribution of amacrine cells.

Response. A previous modeling study by Werner and Passaglia has simulated the OSR with a LN-model with biphasic filter properties. This model did predict a rebound response at the end of the flash sequence, but failed to predict a shift in the latency as a function of the stimulus period with a slope of 1, which we take as a defining feature of the OSR see Discussion 3.1, p.17). In our study,

we also have this control: if we remove the mechanism for short-term plasticity, our model effectively is a LN model, since our filter for individual pathways can be combined to one filter which would lead to an ON- biphasic filter profile. We show the predictions of such a model in our Results 2.4, Figure 4, p.11 and that this is not sufficient to evoke a latency shift with stimulus frequency. We added a sentence to the paragraph to clarify this point in section 2.4, p.9. The novelty of our proposed model is thus short-term plasticity which aids to adapt the response latency to the stimulus. We place this mechanism in the terminal of glycinergic amacrine cells because of our experimental observations.

Remark 7. As mentioned in (6), the OSR is observed when the cell is presented with repeated, high-contrast, non-preferred stimuli. These stimulus statistics are almost certainly never observed in the natural environment. There should be some acknowledgment of this and some discussion of how/whether this behavior would ever be expected in natural vision.

***Response.** We now acknowledge the artificial nature of our experiments in the Discussion, p.16, and point out how we think our results might be relevant for the processing of more natural scenes.*

Remark 8. The manuscript as a whole would benefit from more effort in placing the current work in the context of previous work on the OSR, starting in the Introduction.

***Response.** We elaborate on previous OSR studies in the Introduction, p.2, and set our work in relation to previous studies in the Results section 2.7, p.9. We also want to point out a more extensive comparison in our Discussion 3.1, p.17.*

Minor concerns

Remark 9. In Figure 1B, it appears that there are at least two different modes for the temporal filters (transient vs sustained). How are the authors distinguishing cell type? It seems that there could be several types that show the same behavior.

***Response.** To see if we can identify distinct celltypes amongst ON cells that show an OSR to dark flashes, we performed a clustering on chirp responses of a total of 35 cells that showed a response to this stimulus and had an ON STA, recorded across all experiments. We identified roughly 4 different response types which we classified as "ON-transient", "ON-transient-sustained", "ON-sustained" and "ON-OFF". This suggests that the OSR is not a response behavior specific to a given cell type but rather exhibited by a broader range of cells. We included these results as Supplementary Figure S1 and mention them in Results section 2.1, p.3.*

Reviewer 2 (Remarks to the Author):

This is a very interesting paper and I really enjoyed reading it.

The paper deals with the Omitted Stimulus Response (OSR) in the retina, which is an example of a more general phenomenon in which sensory neurons respond unusually strongly when a repetitive stimulus is interrupted. In other, more general language, the OSR is an example of a “mismatch” signal highlighting the difference between observation and expectation based on the recent history of the sensory input. These types of computations are of general interest in many areas of neuroscience but we understand surprisingly little of the underlying circuit mechanisms.

In this paper the authors home in on the temporal properties of the OSR – the latency between the omitted stimulus and the response, which is dependent on the period of the repeated stimulus. Through a simple but powerful combination of experiment and modelling they provide good evidence for a key insight: depressing inhibitory synapses control the latency of the response to an omitted stimulus. In the retina, these are glycinergic synapses, but the model is quite generic and could be easily transposed to other neural circuits where GABAergic inhibition dominates. I therefore think that this work will have impact not just for the field of retinal processing but also more generally for those investigating the circuit basis of similar “mismatch” computations in other neural circuits. Short-term changes in the strength of both excitatory and inhibitory synapses (primarily depressing changes) are likely common to many (all?) circuits when they are driven hard enough and this paper provides a beautifully approachable example of how this interplay can control the timing of a mismatch response.

The work supports the conclusions well and I cannot see any major flaws in the analysis, modelling or interpretation.

Major points

Remark 1. My major suggestion for improving presentation is to describe more clearly the experimental observations and model predictions for the amplitude of the OSR. These results are important to give an overview of the role of glycinergic inhibition and are briefly mentioned on p. 12 but I would have liked to have seen a more thorough presentation, both because the amplitude of responses is an essential aspect of the phenomenon being modelled and because the current literature is confusing on this point. I suggest that Fig. 1 includes a panel showing how the amplitude of the OSR depends on stimulus period and the effects of glycinergic block on that relation. The model outputs in Fig. 2B indicate that glycinergic block should increase OSR amplitude with a stronger relative effect at lower frequencies. There is a hint of a similar trend in the example experiments in Fig. 1D but a systematic analysis of the whole data set is needed. It may be that the model does not neatly account for changes in amplitude, and this would need discussion.

Response. *In control conditions, the amplitude of the OSR decreases with decreasing frequency in the data, which is captured by our simulations. After*

applying strychnine, the amplitude decreases for high frequencies but increases for low frequencies. This trend is captured by the model as well. However we did not see a significant differences at the population level (see Figure 3). We see a general agreement between model and data on the peak amplitude, but because the trend in the data was not significant, we added these results as Supplementary Figure S3, p.26, and mention them in Results 2.4, p.9.

Figure 3: Peak amplitude to different stimulus frequencies are not systematically impacted by strychnine. **A.** Amplitude of the OSR against stimulus period for control and strychnine conditions in simulations. **B.** Same for experimentally measured amplitudes, not significantly different after Bonferroni-Holm correction (6 Hz $p = 0.26$, 16 Hz: $p = 0.975$).

Remark 2. The abstract says: “Here we show that depressing inhibitory synapses enable the retina to signal an omitted stimulus in a flash sequence”. I don’t think the word “enable” is accurate in terms of what is shown. Blocking glycinergic inhibition does not block the OSR – it adjusts it’s timing. So reword to be more specific and accurate.

Response. We modified this sentence in the Abstract, p.1.

Remark 3. The authors do not investigate the role of GABAergic inhibition and readers outside the retina will wonder why this is not mentioned. Some words at some point about the difficulties of interpreting more generalized block of inhibition across the whole retinal circuit would help those readers understand in which an “omitted experiment response” is generated by the mismatch between their expectations and contents of the paper.

Response. This is an important concern raised by the reviewer and led us to perform additional experiments with Gabazine bath application. We found that the effect of Gabazine on the OSR is quite heterogenous (Figure 4). In some

cells, Gabazine did not affect the OSR (4A), while in others it led to a decrease in slope (4C) and yet in others, it eliminated the peak of the OSR entirely (4E). Our model indeed consists of 2 types of inhibition that serve distinct roles, one (glycinergic) unit that scales the response latency and a (possibly GABAergic) one that generates the OSR via disinhibition. Cells that lose the OSR after Gabazine application are supporting this idea. Nevertheless, the rebound could have other sources than disinhibition such as for example inputs from the OFF pathway or synaptic vesicle accumulation, which would explain why the OSR peak in other cells is not affected. Lastly, it seems that GABAergic inhibition might contribute to the latency scaling for some cells as well. However, due to the variety of effects observed, we decided not to include these results in the manuscript. We mention them in the Discussion 3.2, p.18-19.

Figure 4: Gabazine has various effects on the OSR. **A.,C.,E.** Response traces to flash trains of 3 different frequencies in control (black) and gabazine (shades of purple) conditions. **B.,D.,E.** Mean relative latency to the last flash of the stimulus plotted against stimulus period for control (black) and gabazine (shades of purple) conditions. Dotted line shows slope of 1. **A,B** Example cell where OSR remains largely unaffected by gabazine. (Slope control = 0.71, gabazine = 0.4., $n = 2$) **C,D** Example cell where OSR slope decreases with gabazine. (Slope control = 0.97, slope gabazine = 0.35, $n = 4$). **E,F** Example cell where OSR disappears though gabazine. (Slope control = 0.96, $n = 4$).

Minor points

Remark 4. The abstract says” We also generated new predictions with this model, that we confirmed experimentally.”. It would be good to be more specific about the nature of these predictions (I assume the results in Fig. 5 and 6 are being referred to).

Response. We modified this sentence in the Abstract, p.1.

Remark 5. Section 2.5. para beginning “A characteristic feature in the Omitted Stimulus Response is that the latency shifts by the same amount as the stimulus period. It has been shown that the relative strengths of excitation and inhibition can determine the latency of the response [34, 37, 38, 8].”. At first I thought the OSR in the retina was being referred to but checking the references the authors are discussing the OSR more generally. Would be helpful to add that in first sentence.

Response. We modified this sentence in the Results section 2.4, p.8.

Remark 6. Discussion: sentence beginning “Dynamical synapses have previously been proposed to enable neuronal circuits in the retina to form expectations of future inputs [13] and are thus a plausible candidate to play an important role in the OSR”. The role of excitatory and inhibitory synapses in generating the predictive responses described in ref. 13 (Hosoya et al. 2005) were investigated relatively directly in Johnson et al. 2019 (A Retinal Circuit Generating a Dynamic Predictive Code for Oriented Features. Neuron. 2019 Jun 19;102(6):1211-1222.e3).

Response. We added the suggested reference in the Discussion 3.3, p.19

Remark 7. Discussion could perhaps be improved with some subheadings for major topics.

Response. We add subheadings to the Discussion.

Reviewer 3 (Remarks to the Author):

The manuscript "Temporal pattern recognition in retinal ganglion cells is mediated by dynamical inhibitory synapses" proposes a new mechanism by which the retina may achieve a phenomenon called the Omitted Stimulus Response (OSR). This is a neural behavior previously described in which the retina responds in a special manner to the omission of a stimulus from a sequence of flashes (Schwartz et al. Nature Neuroscience 2007). Here, the authors explore the circuit configuration and properties that could produce this phenomenon. They use standard methods including (1) recording the OSR in the mouse retina using a multi-electrode array to record local potentials of ganglion cells, (2) building a retina model that recapitulates the behavior of the experimentally recorded ganglion cells, and (3) testing whether the model predicts other aspects of how the retina responds to the flashing stimulus. The authors propose an interesting solution, which is that a depressing glycinergic synapse can produce the specific latency features of the OSR—a much less complicated and more biologically plausible solution than those previously proposed in the literature. However, my comments below center on an alternative possibility that has not been addressed and that would re-frame the OSR as a (possibly less interesting) contrast sensing phenomenon. The authors should rule this possibility out.

Overall, these results will be interesting to retinal neurobiologists interested in the role of inhibition in predictive coding the retina.

Major comments

Remark 1. It appears that OSR may be related to a more general phenomenon in the retina, which is the well-described phenomenon of contrast encoding. The dark flashes presented at different frequencies in Fig. 1D are of the same duration, leading to an overall average luminance that is lower during stimulation for the 16 Hz stimulus than during the 6 Hz stimulus. Thus, at the end of this stimulation period, the contrast change is greater for 16 Hz than 6 Hz. It is well-described that bipolar cell release encodes luminance and contrast through mechanisms related to vesicle release at the ribbon and inhibition onto the bipolar cells (ie. Oesch and Diamond 2019, Visual Neuroscience). In addition, the overall stimulation period is different length for each of these stimuli, and it is also established that the duration of suppression of ribbons in both photoreceptors and bipolar cells alters the kinetics and magnitude of their release (i.e. Jackman et al. 2009 Nature Neuroscience; Vlasits et al. 2014 Neuron). The longer that ribbon synapses are suppressed, the larger the glutamate discharge will be upon release from suppression due to an accumulation of vesicles. The fact that the authors do find that they must stimulate with a sufficient number of flashes (Fig 6, 5 vs. 12 flashes) to produce this phenomenon supports the idea that this phenomenon may be related to contrast encoding by ribbon synapses. Thus, I am skeptical of the special/complicated nature of this phenomenon, and wonder whether it is simply a special case of the ganglion cells encoding contrast. This issue does not seem to have been addressed in the

literature on the OSR. To address this, the authors should show that the OSR response is specific to flashing stimuli and that the ganglion cells do not respond in the same manner to a non-flashing stimulus with the same average contrast over time. Another option would be to equalize the contrast change by varying the duration of the flashes in addition to their frequency. The authors should also demonstrate that bipolar cells with luminance/contrast encoding ribbon properties (as described in the literature mentioned above) are not sufficient to produce the OSR phenomenon in their model in the absence of the depressing glycinergic input.

Response. Part 1. *The effect of contrast and luminance of the stimulus is a very interesting point raised by this reviewer, which we address in the first part of this reply. We performed additional experiments which we included in our manuscript to test this and updated our model to explain our new observations. First, it seems important to mention that Schwartz et al., 2007 show in their supplementary material that the OSR to a 12 Hz stimulus is robust to changes in the stimulus pattern. They displayed a sinusoidal variation in light intensity and a dark flash sequence where the mean light level remains constant by increased luminance in interflash intervals and showed the OSR was still present in these cases. However, it is not clear how the latency scaling is affected by these stimulus changes. We therefore performed a set of experiments in which we tested the effect of luminance and contrast modifications on the latency of the OSR. First, we tested whether the duration or the average luminance of the stimulus had an effect on the latency of the offset response (Figure 5). We presented a set of dark step-stimuli of different lengths corresponding to the duration of the sequence presented in the control experiments (5A), and a second set of dark step-stimuli which differed in luminance (5B). Both variations in luminance and variation in durations did not affect the latency of the response after the stimulus end (5C), suggesting that the latency shift in the OSR is specific to periodic stimuli. These results are included in a new section, Results 2.3, Figure 2 A-C. p.6-7.*

Figure 5: *The stimulus duration and contrast of opposite polarity steps do not affect offset-response latency. A. Firing rate responses (lower panel) of one example cell to dark step stimuli with different durations equivalent to the duration of flash sequences of frequencies 6 to 16 Hz. B. Firing rate responses (lower panel) of one example cell to dark step stimuli with different luminances, equivalent to the mean luminance of flash trains with frequencies 6-16 Hz. C. Relation between the response latency and the equivalent stimulus period for step stimuli with duration or luminance modulation.*

We then tested two modifications of our periodic flash trains in which mean luminance was kept constant across stimulus frequencies (Figure 6). The first modification was to keep the luminance steady by increasing light intensity between the dark 40ms flashes, which yielded an OSR with latency shift (shown in blue). This suggests that overall luminance change is neither necessary nor sufficient for the latency shift. In the second modification, we kept the average luminance constant by adjusting the duration of the flashes to half of the stimulus period (in green). Surprisingly, we found that changing the duration of the flashes maintains the OSR peak but erases the latency scaling. These experimental results suggest that the retina selectively detects periodic patterns but can only form temporal predictions for some input patterns. These results are also included in the new section Results 2.3, Figure 2 D-G, p.6-7.

Figure 6: *Latency scaling of the OSR is selective for transient periodic stimuli. A. schematic description of the stimulus modifications to have a constant luminance across frequencies. ‘Duration’ describes a modified stimulus in which the duration of each flash is set to half of the stimulus period. ‘Intensity’ describes a stimulus modification in which the intervals between flashes are of brighter intensity to maintain a constant average luminance level. B. Response traces to flash sequences of 3 different frequencies of each stimulus modification shown in A. C. Relative scaling between response latency and stimulus period for all condition D. Quantification of the fitted slope between peak latency and stimulus period. Control mean 0.89 ± 0.03 , intensity mean 0.74 ± 0.06 , duration mean -0.13 ± 0.01 Stimuli with intensity modulation lead to no significant change in slope ($p = 1.72e - 2$) while duration modulations decrease the slope significantly ($p = 1.2e - 16$).*

Our initially proposed model only partly agreed with these results as it predicted that neither of the two stimulus modifications would yield an OSR with latency shift. We therefore included a new component into our model which is a non-linearity within our intermediate units (see updated panel A in Figure 3/4, Results 2.4, p.8 and Methods 4.2.1 Equation 3 p.22), which can be related to nonlinear processing steps in the inner retina. With this modification, our model predicts the experimentally observed latency scaling, or lack thereof, for all the tested periodic patterns. We updated simulations in Figures 3,4,5,6 and 7 with this new model and show simulations of the new experiments in Supplementary Figure S4, p.27.

Response. Part 2. Another point raised in this remark is that short-term plasticity at excitatory bipolar ribbon synapses could as well account for the latency shift in the OSR. To test this hypothesis, we designed a model based on an ON bipolar cell with a depressing synapse, and tested if this was sufficient to generate an OSR with a latency shift. We found that it was possible to obtain it, but under several conditions. First, the ON bipolar needed a biphasic temporal profile. Second, in the parameters of the dynamical synapse, the release rate needed to be much faster than the recovery rate (70 times faster). In this case, the depressing synapse truncates the rebound response via a fast and strong decrease in vesicle occupancy. This truncation shifts the OSR peak to earlier time-points, and the amount of shifting depends on stimulus frequency: Fast frequencies evoke a larger rebound response and thus an earlier truncation than slow frequencies. This results in a response latency that increases with decreasing stimulus frequency.

A model based on a depressing excitatory synapse could thus predict an OSR, however, such a model cannot explain our experimentally observed strychnine effect. The reviewer mentions an interesting study which shows that presynaptic inhibition can shape the contrast gain in bipolar outputs [Oesch and Diamond, 2019] However, the authors show that this effect is dependent only on GABA_A receptors and not modulated by glycinergic signals. We therefore conclude that, even though depressing synapses in bipolar cells might play a role, the mechanism of depressing inhibitory synapses provides an explanation of the OSR that is more consistent with our experimental data. We included a mention of this in the Discussion 3.3, p.19 and show a simulation of such a model in Supplementary Figure S5, p.28

Figure 7: Model with biphasic ON bipolar cell can simulate the latency shift of the OSR via excitatory plasticity. **A.** Schematic description of the model one biphasic ON unit and synaptic plasticity. **B.** Response traces to 3 stimulus frequencies with the purely excitatory model with plasticity and fixed synaptic strengths. **C.** Scaling between response latency and stimulus period in a model with fixed and dynamic synapses, slope full model 0.94, slope without plasticity 0.44.

Remark 2. While the peak response of the OSR appears to shift with different frequencies of stimulation, the spiking onset appears to begin at about the same time in control and strychnine conditions (Fig. 1D). This suggests that the ganglion cell does start firing at about the same time after the last flash, which is quite different from what is shown in the model (Fig. 2 B) where firing onset and peak both depend on the stimulus frequency in the full model. The authors should explain why peak firing rate is the relevant parameter for calculating the latency to omitted stimulus and why their model results do not match the experimental data in this respect.

Response. The reviewer is right that, in the specific example shown in Fig 1D, the spiking onset seems to begin at the same time for the control and strychnine condition. However, we could not find a consistent trend at the level of the population in our data (Figure 8A). We also could not identify a consistent effect of strychnine (Figure 8B). Additionally, the relation between spiking onset latency and period did not show significant difference between control and strychnine (Figure 8C, $p = 0.0.56$). We therefore conclude that peak firing time-point is a meaningful quantity to look at when modeling the OSR, but not spiking onset. Given the experimental variability of spike onset it is impossible to accurately simulate this quantity. In our simulations, spiking onset scales with the same slope as the peak timing in control simulations (slope = 1.15),

but not in strychnine simulations (slope = 0.36). While this accurately captures the behavior of some cells, it could be possible that other cells (such as the one chosen in our example) receive additional inputs that affect the response start. We added these results in a Supplementary Figure S2, p.26 and mention them in Result 2.2, p.4.

Figure 8: Spiking onset is not a consistent of the OSR latency shift. **A.** Response onset latency against stimulus period in control conditions in all cells that show an OSR in their peak timing. Mean 0.39 ± 0.11 , compared to thick dotted line which has a slope of 1. All latencies are referenced to the latency to the fastest frequency in control condition. **B.** Response onset latency against stimulus period in strychnine conditions in all cells that show a latency shift in their peak timing. Mean 0.15 ± 0.36 , thick dotted line has a slope of 0. All latencies are references to the latency to the fastest frequency in control condition. **C.** Slope quantification, no significant difference $p = 0.056$.

Minor comments

Remark 3. The authors describe the On ganglion cells as biphasic, but in Fig. 1B there appear to be many monophasic cells and a few biphasic cells. The authors should clarify the difference between the cells selected for analysis and other cell types recorded.

Response. We selected cells for the analysis based on two criteria: They have ON polarity as derived from the STA and show an OSR with latency shift slope above 0.7 (see Results section 2.1, p.3 and Methods 4.1.5, p.21). To see if we can identify distinct cell types amongst ON cells that show an OSR to dark flashes, we performed a clustering on chirp responses of a total of 35 cells that showed a response to this stimulus and had an ON STA, recorded across all experiments. We identified roughly 4 different response types which we classified as "ON-transient", "ON-transient-sustained", "ON-sustained" and "ON-OFF". This suggests that the OSR is not a response behavior specific to a given cell type but rather exhibited by a broader range of cells. We included these results as a Supplementary Figure S1. p.26 and mention them in Results 2.1, p.3. It seems like most ON cells with an OSR have a biphasic temporal filter (see panel B in Figure S1), however as this reviewer correctly points out this is not always

the case. We thus removed the characterization as "biphasic" from our headline in Results 2.1 (p.3) and modified our argumentation for model components in Results 2.4 (p.8.) accordingly.

Remark 4. The authors should clarify the units in their modeling plots so it is easier to evaluate whether the values of these parameters are plausible (Fig. 4, 5 voltage and current traces)

Response. We have added the units to voltage and current traces in Figures 4 and 5, (voltages in [mV], currents in [pA] and firing rates in [spikes/s]). As current inputs from each unit to the ganglion cell are simulated with the term $w_X p_\theta(V_X)$, which is in $\frac{mV}{s}$ (from equation 5, p.22 in the manuscript), we calculate currents for visualization purposes in Figures 4 and 5 by dividing this term by the membrane capacitance set to $c_m = 0.1nF$, such that $I_X = \frac{w_X p_\theta(V_X)}{c_m}$. This is now described in the legend of Figure 5, p.14. We also included a scale factor S_{mV} into our model such that all voltage responses are given in mV, starting from the inputs into cellular units (see description after equation 3 in Methods, p. 21).

Remark 5. Blue traces in Fig 5B and Fig. 6C are difficult to distinguish. The authors should make it easier to see the difference.

Response. We changed the color palette for these plots.

Remark 6. The slope of the latency shift appears to become much more variable in the presence of strychnine (Fig. 1G). This reviewer is curious if there are different effects in subpopulations of cells shown here. The authors should demonstrate/describe the details of this distribution by showing the population data with lines between the control and strychnine measurements. In addition, a statistical test that takes into account the unequal variances should be used.

Response. We added the individual data points to Figure 1G, and connected the values for the same cell in control and strychnine conditions with lines. We could not relate this variation to sub-populations of cells in the dataset. We changed the statistical test to use a Welch's t-test to account for the unequal variances (see Figure 1G legend, Methods section 4.1.6, p.21).

Remark 7. The methods section should describe the computer software used for all analysis, modeling, and statistical testing.

Response. We added a section in Methods 4.2.2 (p.23) describing our parameter optimization in more detail and another section in Methods 4.1.6 (p.21) referencing all toolboxes used for optimization and statistical analysis in. The custom code for modeling and analysis will be put online after publication.

Reviewer 4 (Remarks to the Author):

This paper aims at investigating the mechanisms behind the so-called Omitted Stimulus Responses (OSR) in retinal ganglion cells. This previously observed phenomenon describes the strong response peak in some ganglion cells when a sequence of periodic light stimuli suddenly ends or misses an individual stimulus in the sequence. In particular, when the period of the light stimulus sequence increases, the latency of the response peak increases in a corresponding fashion.

In the present manuscript, the authors combine experimental recordings from mouse retina and computational analyses to derive a circuit model that aims at explaining the OSR and its latency dependence on the stimulus period. As a key element, the authors propose that ON ganglion cells receive OFF inhibition via a synapse with short-term depression. This is an interesting model that is nicely illustrated and analyzed.

Remark 1. However, a substantial drawback of the presented material is that the investigated scenario does not correspond to the typical phenomenon of an OSR. The authors investigate responses of ON cells to a sequence of OFF flashes, whereas the classical investigations of the OSR in the retina focused on responses of OFF cells to such a sequence of OFF flashes (even though some examples of ON cells were also included). The fundamental difference is that, for an ON cell, the termination of the OFF flashes corresponds effectively to an increase in luminance and thus an activating stimulus. Adding a latency dependence to the resulting response seems not so mysterious and has not been, for example, the issue of interest in previous investigations of the mechanism of the OSR. For the classical OSR, on the other hand, one would need to explain how the response peak of an OFF cell, after cessation of OFF-type stimulation, is generated in the first place, in conjunction with its latency dependence on stimulus frequency.

Response. *We focused on the response of ON cells to a series of dark flashes, and on how the latency shifts with the period of the stimulus, for several reasons: -A previous study by Schwartz et al.2008, has shown that the ON pathway, which can be blocked pharmacologically with L-AP4, is necessary to obtain an Omitted Stimulus Response to dark flashes. -In a paper by Werner et al. 2008, they claim that one could model the OSR as a rebound response coming from the ON pathway, and that additional responses to dark flashes during the sequence could be modeled by a convergence of an ON and an OFF pathway onto the same ganglion cell, with OFF pathway responsible for the flashes during the sequence, and the ON pathway responsible for the OSR. -However, the model from Werner et al did not reproduce the experimental result of a scaling of peak latency with the period of the stimulus with a slope of 1 and acts like our model without short-term plasticity. We clarify this in Results section 2.4, p. 9 and Discussion section 3.1, p.17. For this reason we decided to focus on ON ganglion cell responses to a dark flash sequence, and on the scaling of latency with period, because this is the part that was not explained by these previous studies. We agree with the reviewer that our model as it is does not explain how*

OFF ganglion cells would respond to both the sequence and present an OSR, but, following Werner et al, we think that this could be obtained by adding another independent OFF pathway that would deliver an input to the same ganglion cell. This pathway would trigger a response during the flashes, while the circuit described in our model would generate an OSR that scales with latency. We have added these points in the Discussion section 3.2, p.18.

Remark 2. In the present model, for example, the response seems not to depend on the periodic nature of the stimulus, and a continuous dimming would probably cause similar rebound activation, with latency likely depending on the amplitude of the dimming stimulus.

Response. We tested experimentally by presenting a set of non-flashing dark step-stimuli with different levels of dimming (Figure 9). The rebound response to these stimuli had the same latency, suggesting that the latency shift in the OSR is specific to periodic stimuli, and cannot be obtained by a continuous dimming.

Figure 9: Luminance dimming does not affect the latency of the rebound response. **A.** Firing rate responses (lower panel) of one example cell to dark step stimuli with different light intensities equivalent to the mean light intensity of flash trains with frequencies 6-16 Hz. **B.** Scaling between response latency and corresponding stimulus period for step stimuli luminance modulation. x-axis in % of maximal luminance of the dimmed step. Baseline is at 50%.

Remark 3. The implementation of the rebound is also somewhat confusing, as the manuscript states that the voltages that form the input into the ganglion cell undergo a thresholding operation (Equation 5). Yet, the thresholds for the ON excitation and ON inhibition are not given in the table, and it seems that these inputs should be mostly linear (or have a very low threshold) for the rebound to occur. This is not impossible, but less plausible than rectifying synapses, and effectively, the ON inhibition then acts like an OFF excitation.

Response. In our previous version we indeed only rectified the adapting OFF unit of our model, while ON inputs were linearly fed into the ganglion cell. As the reviewer correctly describes, both ON excitation and inhibition need to operate in the linear regime in order to have an OSR peak, which arises from

hyperpolarization in the ON inhibitory pathway. In our experiments, the baseline illumination is a gray background. Thus, before the flash sequence starts, we assume our cells to be at an adapted voltage level above threshold from which they can either depolarize or hyperpolarize. For more plausibility and consistency, we included rectification at all synapses with low thresholds in the ON components (see Results 2.4 (p.8), Methods 4.2.1 (p.22) Equation 5 and Table 1 (p.23)). Nevertheless, it is certainly possible that the rebound comes from other sources than ON inhibition, such as OFF excitatory input, or dynamics in the bipolar ribbon synapse where vesicles accumulate.

Remark 4. In addition, while the modeling of the depressing inhibitory input is interesting, the evidence of such a mechanism in the experimental data is not very strong. The block of glycinergic inhibition is a rather unspecific test, which likely throws off the excitation-inhibition balance and could thus disturb the latency dependence even if this inhibition is not directly involved in the mechanism. Also, the derived predictions (e.g. lower OSR amplitude for lower frequencies and shorter sequences) are not very specific for the depressing inhibitory input, as the predictions essentially state that a weaker OFF stimulus leads to weaker rebound response. This can likely also be explained by nonlinear or depressing excitatory input.

Response. Following the suggestions of reviewers, we have done several other experiments whose outcome is predicted by our model, see new Results 2.3 (p.6-7) for experiments and Results 2.4 (p.9) and new Supplementary Figure S4 (p.27) for simulations. This shows that our model can explain a broader range of experiments. We have also tested if a model based on a depressing excitatory synapse could explain our results. We found that it was possible to predict an OSR with latency shift with a depressing excitatory synapse model (see Figure 10), but under several conditions. First, the ON bipolar needed a biphasic temporal profile. Second, in the parameters of the dynamical synapse, the release rate needed to be much faster than the recovery rate (70 times faster). In this case, the depressing synapse truncates the rebound response via a fast and strong decrease in vesicle occupancy. This truncation shifts the OSR peak to earlier time-points, and the amount of shifting depends on stimulus frequency: Fast frequencies evoke a larger rebound response and thus an earlier truncation than slow frequencies,. This results in a response latency that increases with decreasing stimulus frequency. A model based on a depressing excitatory synapse could thus predict an OSR, however, such a model cannot explain our experimentally observed strychnine effect. We therefore conclude that, even though depressing synapses in bipolar cells might play a role, the mechanism of depressing inhibitory synapses provides an explanation of the OSR that is more consistent with our experimental data. We included a mention of this in the discussion section 3.3, p.19 and show a simulation of such a model in Supplementary Figure S5 (p.28).

Figure 10: Model with biphasic ON bipolar cell can simulate the latency shift of the OSR via excitatory plasticity. **A**. Schematic description of the model one biphasic ON unit and synaptic plasticity. **B**. Response traces to 3 stimulus frequencies with the purely excitatory model with plasticity and fixed synaptic strengths. **C**. Scaling between response latency and stimulus period in a model with fixed and dynamic synapses, slope full model 0.94, slope without plasticity 0.44.

REVIEWERS' COMMENTS

Reviewer #1 (Remarks to the Author):

The authors have adequately addressed my previous concerns.

Reviewer #2 (Remarks to the Author):

The authors have carefully and thoughtfully responded to all my points, including further results on changes in the amplitude of the OSR. It is a pity that experiments using GABA_A block did not yield a simple pattern of results, but that might not be surprising.

I think that the revised manuscript is significantly improved and reports a very interesting study.

Reviewer #3 (Remarks to the Author):

The authors have addressed all of my concerns and the results are now much stronger. This manuscript now convincingly describes a mechanism for error signaling using depressing inhibition, which will be of broad interest to neuroscientists.

Reviewer #4 (Remarks to the Author):

The manuscript has been much improved. In particular, the additional experiments with variations of the flash sequence and sustained stimuli at different contrast and duration provide important controls. Also, the discussion of whether depressing excitatory input could provide an alternative explanation is a welcome addition. However, I do have a few remaining comments.

I maintain that it is a bit confusing and perhaps misleading to focus the analysis on ON cell to a sequence of dark flashes, whereas previous work that made the Omitted Stimulus Response (OSR) popular in the retina, was focused on OFF cells, where the occurrence of a post-sequence response is arguably more puzzling. It would be helpful to point out this particular focus and deviation from previous studies early on (and not only late in the Discussion).

Similarly, while the authors speak of an OSR, they never test an actual omission of flash stimulus, only the cessation of the sequence. Given the often long latencies of the response, it may well be that for an omitted flash, the subsequent stimulus flashes may interfere with the response. Again, this discrepancy of terminology and investigation should at least be spelled out clearly and discussed.

A second point is that some early statements seem a bit overly strong regarding the significance of the observation of an OSR in light of the fact that the analysis selects a small subset of recorded cells on the condition that they show an OSR with the desired latency shift (40 out of 143 cells that showed a response after stimulus offset out of a perhaps larger number of totally recorded ON cells). For example, the statement that the retina would have a precise temporal expectation of the expected stimulus and adjusts its responses accordingly (lines 79ff) is doubtful and almost circular given the cell selection based on exactly this property. In view of this, it would actually be interesting to know what the other 73% of cells with a response after offset of the flash sequence do under, e.g., strychnine application. Do some of these now have an increase in the slope of the latency versus period relationship? Maybe the authors can at least make it a bit clearer how the stringent cell selection

affects their analyses and interpretations.

Minor points:

The experiment with the increase light intensity between the flashes (Fig. 2D-G) is a nice control. It's a bit puzzling, though, that the ON cells (at least in the example) don't respond to the transient increase. Is this a general observation? If not, does the activity during the flash sequence affect the post-sequence response?

Fig. 2A is missing axis labels; x axis is presumably in sec? And Fig. 2F gives units of ms on the axis, but the values (up to 0.08) rather suggest sec.

Second Response to Reviewer Comments

Nature Communications manuscript NCOMMS-23-06244-T

March 2024

First of all, we would like to thank all reviewers again for their constructive feedback and insightful remarks.

To comply with nature communications submission guidelines, we included all figures used in reviewer replies as supplementary figures in the manuscript and added a schematic summary figure of our main findings in the Discussion. We also moved all figures to the end of the manuscript.

We included a total of 3 more figures in our manuscript:

- supplementary figure 2 : density of absolute latencies of OSRs to all frequencies
- supplementary figure 7 : preliminary results on GABAzine effects on the OSR
- main text figure 8: graphical summary

As before, all remarks from the reviewer are stated below and followed by a response to each remark (in *italic*). In response to remaining remarks, we modified our text at several points in Results and Discussion. All changes in the manuscript are highlighted in red and referenced with a tag to the respective remark labelled [RnPk], where n is the reviewer index and k the remark index.

Reviewer 1 (Remarks to the Author):

The authors have adequately addressed my previous concerns.

Reviewer 2 (Remarks to the Author):

The authors have carefully and thoughtfully responded to all my points, including further results on changes in the amplitude of the OSR. It is a pity that experiments using GABAzine block did not yield a simple pattern of results, but that might not be surprising.

I think that the revised manuscript is significantly improved and reports a very interesting study.

Reviewer 3 (Remarks to the Author):

The authors have addressed all of my concerns and the results are now much stronger. This manuscript now convincingly describes a mechanism for error signaling using depressing inhibition, which will be of broad interest to neuroscientists.

Reviewer 4 (Remarks to the Author):

The manuscript has been much improved. In particular, the additional experiments with variations of the flash sequence and sustained stimuli at different contrast and duration provide important controls. Also, the discussion of whether depressing excitatory input could provide an alternative explanation is a welcome addition. However, I do have a few remaining comments.

Remark 1. I maintain that it is a bit confusing and perhaps misleading to focus the analysis on ON cell to a sequence of dark flashes, whereas previous work that made the Omitted Stimulus Response (OSR) popular in the retina, was focused on OFF cells, where the occurrence of a post-sequence response is arguably more puzzling. It would be helpful to point out this particular focus and deviation from previous studies early on (and not only late in the Discussion).

Response. We now point out in Results Section 2.1 (p.3) that the OSR is a phenomenon in both ON and OFF cells, and justify our focus on ON cells via previous pharmacological experiments that show the OSR disappearing under block of ON bipolar cell transmission (Schwartz et al., 2007).

Remark 2. Similarly, while the authors speak of an OSR, they never test an actual omission of flash stimulus, only the cessation of the sequence. Given the often long latencies of the response, it may well be that for an omitted flash, the subsequent stimulus flashes may interfere with the response. Again, this discrepancy of terminology and investigation should at least be spelled out clearly and discussed.

Response. We now mention in Discussion 3.2, p.11, that we focused on the OSR as the response to the abrupt cessation of a periodic stimulus, as is has been the case in a number of previous studies (Schwartz et al., 2007 and 2008, Werner et al. 2008, Gao et al. 2009, Tanaka et al. 2019). We also acknowledge that it has been shown that retinal ganglion cells in both mouse and salamander emit an OSR to more complex pattern violations, for example in response to an omission in the middle of a periodic stimulus sequence (Schwartz et al., 2007), which we did not test here. Such a response to an omission would be guaranteed if the overall latency of the OSR is shorter than twice the stimulus period to allow for the OSR, to occur before the flash sequence continues after omission. According to absolute latencies of cells recorded in this study, several cells should signal omissions in the middle of a flash stimulus up to frequencies of 12 Hz.

We included a supplementary Figure with absolute latencies to all frequencies tested, compared to $2T$ for each period $T = \frac{1}{f}$:

Figure 1: **Absolute latency of the OSR after all frequency tested of $n = 35$ cells included in our study.** Shaded colored areas shows the density, individual datapoints are shown in light blue. Dotted black line indicates $2T$ for each period $T = \frac{1}{f}$. 6 Hz: max = 615 ms , min 216 ms, median = 340 ms; 8 Hz: max = 547 ms , min 189 ms, median = 232 ms; 10 Hz: max = 529 ms , min 178 ms, median = 220 ms; 12 Hz: max = 497 ms , min 159 ms, median = 207 ms; 16 Hz: max = 483 ms , min 142 ms, median = 188 ms;

Remark 3. A second point is that some early statements seem a bit overly strong regarding the significance of the observation of an OSR in light of the fact that the analysis selects a small subset of recorded cells on the condition that they show an OSR with the desired latency shift (40 out of 143 cells that showed a response after stimulus offset out of a perhaps larger number of totally recorded ON cells). For example, the statement that the retina would have a precise temporal expectation of the expected stimulus and adjusts its responses accordingly (lines 79ff) is doubtful and almost circular given the cell selection based on exactly this property. In view of this, it would actually be interesting to know what the other 73% of cells with a response after offset of the flash sequence do under, e.g., strychnine application. Do some of these now have

an increase in the slope of the latency versus period relationship? Maybe the authors can at least make it a bit clearer how the stringent cell selection affects their analyses and interpretations.

Response. *We changed our formulation in lines Results 2.1, p.3 line 79ff to specify that not the whole retina but only a small subset could carry temporal expectations. We also emphasize now in Results 2.2 (p.4) that we performed the presented analysis on the effect of strychnine on ON retinal ganglion cell responses which exhibited a OSR with latency shift in control conditions. Within the remaining 70% of cells that responded after the stimulus end but not shift their latency with the period, the effect was very heterogeneous. Some responses were almost unchanged, while others disappeared or changed their latency, albeit in a less systematic way.*

Minor points:

Remark 4. The experiment with the increase light intensity between the flashes (Fig. 2D-G) is a nice control. It's a bit puzzling, though, that the ON cells (at least in the example) don't respond to the transient increase. Is this a general observation? If not, does the activity during the flash sequence affect the post-sequence response?

Response. *We confirm that the responses to the stimulus with increased light intensity between the flashes were almost identical to the responses in the control stimulus. To both the control stimulus and the modification, only 4 out of 20 ON cells with OSR showed a response during the stimulus, mostly for low frequencies. We did not identify any quantifiable differences in the OSR response in these two cells compared to the remaining 16 cells that did not respond during the stimulus. Our hypothesis is that the transient increases are either too fast (for high frequencies) or too small (for low frequencies), to evoke a response above threshold. We mention this in results 2.3, p5.*

Remark 5. Fig. 2A is missing axis labels; x axis is presumably in sec? And Fig. 2F gives units of ms on the axis, but the values (up to 0.08) rather suggest sec.

Response. *We added the x-axis label to Figure 2F and changed the scale of the latency in Figure 2F to milliseconds, which was indeed in seconds before.*